# WHEN GREEDY WINS: EMERGENT EXPLOITATION BIAS IN META-BANDIT LLM TRAINING

**Sanxing Chen[1], Xiaoyin Chen[2,3], Yukun Huang[1], Roy Xie[1], Bhuwan Dhingra[1]**
[1]Duke University, [2]Mila - Québec AI Institute, [3]Université de Montréal
`sanxing.chen@duke.edu, bdhingra@cs.duke.edu`

## ABSTRACT

While Large Language Models (LLMs) hold promise to become autonomous agents, they often explore suboptimally in sequential decision-making. Recent work has sought to enhance this capability via supervised fine-tuning (SFT) or reinforcement learning (RL), improving regret on the classic multi-armed bandit task. However, it remains unclear how these learning methods shape exploration strategies and how well they generalize. We investigate both paradigms by training LLMs with SFT on expert trajectories and RL with a range of tailored reward signals including a strategic, regret-shaped reward to reduce variance, and an algorithmic reward that enables oracle imitation. The resulting agents outperform pre-trained models and achieve performance comparable to Upper Confidence Bound (UCB) and Thompson Sampling, with robust generalization to $6\times$ longer horizons and across bandit families. Behavioral analysis reveals that gains often stem from more sophisticated but greedier exploitation: RL/SFT agents are more prone to early catastrophic failure than pre-trained models, prematurely abandoning exploration. Furthermore, agents trained to imitate UCB learn to outperform their teacher by adopting more exploitative variants. Our findings clarify when each training paradigm is preferable and advocate tailored reward design and evaluation beyond average regret to promote robust exploratory behavior. [1]

## 1 INTRODUCTION

A fundamental challenge in sequential decision-making problems lies in the exploration-exploitation trade-off, where an agent must balance exploiting known good actions with exploring new ones to discover potentially better options. The multi-armed bandit (MAB) problem serves as a classic, formalized testbed for studying this critical behavior. Despite their sophisticated capabilities, Large Language Models (LLMs) often struggle here, defaulting to short-sighted, greedy behavior that over-exploits known rewards at the expense of exploration (Krishnamurthy et al., 2024; Schmied et al., 2025). While certain prompting configurations can elicit better performance from frontier models like GPT-4, this inherent suboptimal bias remains a significant hurdle for most models.

To address this, two primary training paradigms have emerged for shaping LLM exploration behavior: Supervised Fine-Tuning (SFT) and RL. SFT teaches the LLM to mimic the behavior of an optimal exploration algorithm, such as Upper Confidence Bound (UCB), by training on trajectories of expert demonstrations. In contrast, RL enables the model to learn an effective policy directly from environmental rewards. When trained to solve bandit instances that differ from those they encountered during training, LLMs effectively become meta-bandit agents, acquiring meta-policy capable of exploring novel environments (Kveton et al., 2020). Prior works suggest that both methods can improve exploration capabilities in LLMs on in-distribution tasks, with SFT showing more consistent results (Nie et al., 2024; Schmied et al., 2025). However, a deeper understanding of how these training methods shape an agent's strategy is lacking. It is unclear whether the policies induced by SFT and RL differ mechanistically. More critically, how do these policies generalize to longer horizons and out-of-distribution environments?

In this work, we train LLMs to perform MAB tasks using both SFT on expert trajectories and RL with a spectrum of task-specific reward signals. We evaluate the performance of learned policies

---

[1]Our code and data are released in `https://github.com/sanxing-chen/meta-bandit-llm`.

on a range of MAB environments, under length generalization and cross-distribution transfer (e.g., Gaussian to Bernoulli). In addition to the standard stochastic reward of bandits, we propose two additional reward signals: a **strategic reward** based on the notion of regret to reduce training variance, and an **algorithmic reward**, which incentivizes imitation learning of an oracle policy like UCB via RL. We find that both SFT and RL improve the base model's performance on MAB tasks in achieving lower regret and higher rewards, achieving comparable performance to theoretical optimal baselines like UCB and Thompson Sampling. For RL, the strategic reward improves training efficiency in high-variance environments, while the algorithmic reward consistently outperforms other learned policies due to the ease of credit assignment. Moreover, RL policies yield more robust generalization across different bandit families compared to SFT. The policies also exhibit strong generalization on $6\times$ longer (compared to training) and out-of-distribution environments.

While achieving lower regret is the canonical measure of success in MAB, classical literature cautions that relying solely on this aggregate statistic can obscure important characteristics of the agent's behavior (Lattimore & Szepesvári, 2020). An agent might achieve a superior average performance with a high-risk policy prone to catastrophic failure, a nuance that the expected outcome can overlook. This prompts a deeper question: does a lower average regret achieved by the LLM policies indicate the acquisition of a robust exploration strategy?

To answer this question, we analyze the agents' action patterns and compare them to pre-trained models and baselines like UCB and Greedy policies. We utilize surrogate statistics such as suffix failure rate, which is highly suggestive of the long-term prospects of the agent (Krishnamurthy et al., 2024). We find that the agents' impressive improvements in performance are linked to learning more sophisticated forms of exploitative behavior. For instance, agents trained via RL to imitate an optimal UCB policy often outperform their teacher by implementing variants of UCB that can prematurely stop exploring an action after unsatisfactory short-term rewards. This suggests that while the training process maximizes average performance with reasonable generalization, it incentivizes short-term reward seeking that can be counterproductive in the long run. The suitability of these learned policies ultimately depends on whether an application prioritizes long-term robustness over immediate returns, or average performance over worst-case scenarios.

In summary, we present a unified study of how SFT and RL shape LLM exploration in MAB, treating trained models as meta-bandit agents. We introduce two principled reward designs—strategic (regret-shaped) rewards that stabilize learning in high-variance settings and algorithmic rewards that enable efficient RL-based imitation of oracle policies, which both improve over baselines from prior work (Schmied et al., 2025), with algorithmic rewards yielding the most consistent gains. Evaluations demonstrate robust generalization to $6\times$ longer horizons and across bandit families, with RL policies transferring more reliably than SFT. Beyond aggregate regret, our behavioral analysis reveals mechanistic differences: learned policies often implement exploitative strategies that boost average returns but can sacrifice long-term robustness.

## 2 RELATED WORK

The multi-armed bandit problem, despite being a classical abstraction, embodies the fundamental exploration-exploitation trade-off central to sequential decision-making and has wide real-world applications (Bouneffouf et al., 2020; Bouneffouf & Feraud, 2025). As LLMs are increasingly deployed in interactive settings, the MAB problem has become a key testbed for evaluating their ability to incrementally gather information and improve over time, a paradigm known as In-Context Reinforcement Learning (ICRL) (Moeini et al., 2025).

Bandit problems have long been used to evaluate the generalizable ICRL capabilities of sequential models like RNNs and Transformers (Duan et al., 2016; Laskin et al., 2023; Lee et al., 2023). In the LLM era, initial benchmarks found that pre-trained models can learn to explore simple MAB problems in-context (Binz & Schulz, 2022; Wu et al., 2024; Coda-Forno et al., 2023; Park et al., 2025). However, they exhibit unsatisfactory exploratory behavior in complex environments without careful prompt engineering (Krishnamurthy et al., 2024; Monea et al., 2024). Subsequent work has sought to address this through activation steering (Rahn et al., 2024) and fine-tuning (Tajwar et al., 2025). Nie et al. (2024) uses supervised fine-tuning (SFT) on expert trajectories to improve performance, demonstrating successful generalization to different reward distributions within the

same bandit class. More recently, Schmied et al. (2025) applies reinforcement learning to train LLMs for bandit tasks, showing positive but weaker in-distribution results compared to SFT.

Our work provides a systematic comparison of these two learning paradigms. We demonstrate that RL-trained agents, while matching SFT performance in-distribution, generalize more effectively to out-of-distribution environments. More importantly, we move beyond simple performance comparisons to conduct a behavioral analysis that uncovers subtle but critical failure modes in how LLMs learn to explore, highlighting previously unaddressed challenges.

## 3 METHODOLOGY

A MAB problem $\mathcal{B} = (\mathcal{A}, R)$ is defined as a set of arms $\mathcal{A} = \{1, \ldots, K\}$, where each arm $i \in \mathcal{A}$ is associated with a reward distribution $R_i$ and mean $\mu_i$. The goal of the agent is to maximize the expected cumulative reward $\mathbb{E}[\sum_{t=1}^{T} r_t]$ over $T$ trials. During training, the agent learns from bandit instances sampled from an unknown task distribution $\mathcal{D}$. We can evaluate the learning agent's performance *in-distribution* by sampling bandit instances from $\mathcal{D}$ or *out-of-distribution* (OOD) on instances from a different distribution $\mathcal{D}'$. In training an agent to solve various bandit instances from a task distribution, we are effectively searching for a **meta-bandit** policy (Kveton et al., 2020), which is a reinforcement learning problem.

> **Prompt with summary statistics**
>
> In a 5-armed bandit problem, here are the results of previous arm pulls:
>
> Arm 0: 1 pull, avg. reward -0.249
> Arm 1: 2 pulls, avg. reward 0.281
> Arm 2: 7 pulls, avg. reward 0.790
> Arm 3: 3 pulls, avg. reward 0.279
> Arm 4: 7 pulls, avg. reward 1.015
>
> Which arm should be pulled next? Show your reasoning in <think> </think> tags and your final answer in <answer> </answer> tags.

Figure 1: An instruction provided to the LLM agent for the MAB task.

### 3.1 REINFORCEMENT LEARNING OF META-BANDIT LLM AGENTS

At each bandit turn $t$, the LLM agent takes as input the interaction history consisting of past actions and rewards in the observation $o_t$, and generates a sequence of tokens $s_t$ which contains the action of the next arm to pull $a_t$. The environment then returns the stochastic reward $r_t \sim R_{a_t}$. The interaction history is then updated with $o_{t+1} = f(o_t, a_t, r_t)$, where $f$ can be a simple concatenation or, in our case, a summarizer that extracts sufficient statistics as shown in Figure 1. The process is repeated for $T$ turns for each episode. As the agent learns over a history to build its belief about the environment (e.g., distribution family and variance), this process forms a Partially Observable Markov Decision Process (POMDP). It can be trained using on-policy RL to maximize episodic return and thus learns an amortized exploration strategy over histories.

Unlike traditional RL policies that directly select actions, LLM agents operate in the token space. This implementation converts the problem into a two-level hierarchical MDP (Hauskrecht et al., 2013; Xue et al., 2025), where a high-level policy operates at the turn level to select a local policy that generates the entire response $s_t$ and receives the external reward $r_t$. The low-level policy operates at the token level to implement the selected local policy. The probability of generating the token $s_{t,j}$ at position $j$ is given by: $\pi_\theta(s_{t,j}|o_t, s_{t,<j})$ where $s_{t,<j}$ is the sequence of tokens generated in turn $t$ up to position $j-1$. At turn $t$, the token index $j$ ranges from $J_{t,\text{start}} = |o_t| + 1$ to $J_{t,\text{end}} = |o_t| + |s_t|$. We pass $r_t$ as the reward signal to the low-level policy at $J_{t,\text{end}}$, while there is no reward signal for intermediate tokens.

To learn $\pi_\theta$, we adopt PPO (Schulman et al., 2017) and compute token-level advantages with a dual-$(\gamma, \lambda)$ Generalized Advantage Estimator (Schulman et al., 2016). We use separate discount factors and trace-decay coefficients for intra-turn and inter-turn steps, denoted $\gamma_{\text{intra}}$, $\gamma_{\text{inter}}$ and $\lambda_{\text{intra}}$, $\lambda_{\text{inter}}$, respectively. For simplicity, we define the token-level state at step $j$ as $h_{t,j} = (o_t, s_{t,<j})$. The one-step temporal difference (TD) error for each generated token index $j$ is:

$$\delta_{t,j} = \begin{cases} \gamma_{\text{intra}} V(h_{t,j+1}) - V(h_{t,j}) & \text{if } J_{t,\text{start}} \leq j < J_{t,\text{end}} \\ r_t + \gamma_{\text{inter}} V(o_{t+1}) - V(h_{t,j}) & \text{if } j = J_{t,\text{end}} \end{cases} \tag{1}$$

For the final token at index $J_{t,\text{end}}$, the error incorporates the external reward $r_t$ and bootstraps from the value of the next turn's initial state, $V(o_{t+1})$, using the inter-turn discount factor $\gamma_{\text{inter}}$. In practice, since we can only optimize over a truncated horizon for this infinite-horizon problem, we infer the value of one more turn, $V(o_{T+1})$ for the last turn $T$.

The GAE advantage for token index $j$ now accumulates TD errors over all subsequent generated-token positions across the entire episode. Let $\kappa(\tau)$ denote the starting generated-token index in turn $\tau$ as seen from $(t, j)$: $\kappa(\tau) = \begin{cases} j & \text{if } \tau = t \\ J_{\tau,\text{start}} & \text{if } \tau > t \end{cases}$. Define the step-weighting product from $(t, j)$ to $(\tau, k)$ as:

$$P(t, j, \tau, k) = \left[ \prod_{p=t}^{\tau-1} (\lambda_{\text{inter}} \gamma_{\text{inter}}) \left( \prod_{u=\kappa(p)}^{J_{p,\text{end}}-1} \lambda_{\text{intra}} \gamma_{\text{intra}} \right) \right] \left( \prod_{u=\kappa(\tau)}^{k-1} \lambda_{\text{intra}} \gamma_{\text{intra}} \right).$$

The token-level GAE advantage for $(t, j)$ is then:

$$\hat{A}_{t,j} = \sum_{\tau=t}^{T} \sum_{k=\kappa(\tau)}^{J_{\tau,\text{end}}} P(t, j, \tau, k) \, \delta_{\tau,k}.$$

With token-level advantages defined only for generated tokens, the clipped PPO objective is:

$$\mathcal{L}^{\text{PPO}}(\theta) = \hat{\mathbb{E}}_{t,j} \left[ \min \left( r_{t,j}(\theta) \, \hat{A}_{t,j}, \; \text{clip}(r_{t,j}(\theta), 1 - \epsilon, 1 + \epsilon) \, \hat{A}_{t,j} \right) \right],$$

where the per-token probability ratio is $r_{t,j}(\theta) = \frac{\pi_\theta(s_{t,j}|h_{t,j})}{\pi_{\theta_{\text{old}}}(s_{t,j}|h_{t,j})}$. Here $\theta_{\text{old}}$ is the reference policy parameter at the previous iteration. This objective trains the policy at token level using the two-scale GAE that respects intra-turn and inter-turn dynamics. We intentionally omit the KL-divergence term, which is often employed in PPO as we find it to be unnecessary for our setting without a learned reward model.

## 3.2 REWARD DESIGN

As described above, the meta-bandit agent relies solely on the past interaction history $o_t$ to generate the next action. The interaction history $o_t$ is a summary of the past actions and rewards, which is tied to the stochastic bandit rewards $r_t$ and cannot be changed. We can however opt for a different reward signal for the PPO optimization in Equation 1. The **original bandit rewards** (RL-OG), although a natural choice of reward signal for PPO optimization, contribute to credit assignment difficulty and learning inefficiency due to their intrinsic stochasticity.

On the other hand, we can more accurately measure the optimality of an action based on the notion of immediate regret. At each time step, the immediate regret is defined as the difference between the expected reward of the optimal arm and the expected reward of the arm selected by the agent. $\Delta_t = \mu^* - \mu_{A_t}$. We define the **strategic reward** (RL-STG) based on the immediate regret of the agent's action:

$$\tilde{r}_t = 1 - \frac{\Delta_t}{\Delta_{\max}} = \frac{\mu_{A_t} - \min_i \mu_i}{\mu^* - \min_i \mu_i} \in [0, 1].$$

This reward signal directly optimizes an action's utility, which simplifies credit assignment. Using the realized regret as the reward is a form of baseline subtraction (Kveton et al., 2020). We further use the (pseudo) regret, which is analogous to introducing a control variate. While this approach reduces variance, it theoretically does not alter the optimal policy to which the agent converges.

A third approach moves beyond extrinsic environmental rewards, instead using a reward function optimized by an expert oracle (Ciosek, 2022). While this reward function often needs to be learned via inverse reinforcement learning (Abbeel & Ng, 2004), we can bypass this by deriving it directly from well-established optimal algorithms for MAB. We select the Upper Confidence Bound (UCB) algorithm as the oracle policy, as its deterministic and distribution-agnostic properties provide a consistent and unambiguous learning signal. We therefore define the **algorithmic reward** (RL-ALG) as a binary signal $r_t = 1$ if the agent's action matches the oracle's decision $\pi_{\text{oracle}}(o_t)$, and $r_t = 0$ otherwise. Because the UCB oracle is a reactive algorithm, this myopic reward is sufficient for on-policy learning and sidesteps the need for return-based credit assignment. This imitation learning setup leaves the agent free to discover its own internal algorithm for processing the interaction history to match the oracle's choice at each step, without any supervision on the reasoning process.

> **Demonstration of UCB calculation**
>
> <think> Let me calculate the UCB value for each arm after $(1 + 2 + 7 + 3 + 7) = 20$ pulls:
>
> Arm 0: Uncertainty bonus $= \sqrt{\ln(20)/1} \approx \sqrt{2.996/1} \approx 1.731$; UCB $= -0.249 + 1/2 \times 1.731 = 0.616$
>
> *Details of Arm 1-3 omitted for brevity.*
>
> Arm 4: Uncertainty bonus $= \sqrt{\ln(20)/7} \approx \sqrt{2.996/7} \approx 0.654$; UCB $= 1.015 + 1/2 \times 0.654 = 1.342$
>
> Based on these calculations, I choose arm 4 as it has the highest UCB value. </think>
> <answer> Arm 4 </answer>

Figure 2: An example of the UCB calculations for the state in Figure 1, used in SFT.

On top of these task specific rewards, we also consider a reward shaping term that encourages the LLM agent to generate valid responses. Specifically, we set reward to zero if our parser cannot extract a valid action and rationale from the response. For the stochastic reward setting (`RL-OG`), because the unbounded reward is sometimes negative, we subtract $0.5$ from the reward as the penalty for invalid responses.

### 3.3 SUPERVISED LEARNING

We also consider a supervised fine-tuning (`SFT`) baseline, where the LLM agent is trained on observation-response pairs. The response includes synthetic CoT demonstrations to explicitly calculate UCB values and the UCB action (Figure 2). Here, both the rationales and the actions are directly supervised. Since the states embodied by the observation are sampled from the UCB policy, the learning process is off-policy.

## 4 EXPERIMENTAL SETUP

**Language Model Configuration.** We use Qwen 2.5 3B and 7B Instruct (Qwen et al., 2024) as the base model for fine-tuning. The observation at each time step consists of a natural language instruction of the MAB task and the interaction history presented as a summary of the number of pulls and average reward for each arm (Figure 1). We use this sufficient statistics to summarize the interaction history, which has been shown to be more effective than using a cumulative context, e.g., a raw list of actions and rewards (Krishnamurthy et al., 2024). In the instruction, the agent is asked to think step-by-step using chain-of-thought reasoning, which is critical for eliciting the sequential decision-making ability of LLMs (Yao et al., 2023).

**RL Configuration.** We build our RL training code on top of the VeRL framework (Sheng et al., 2024). At each training iteration, we first sample a batch of 64 random environments from the training task distribution $\mathcal{D}$. From each environment, we collect a rollout of length $T = 50$, resulting in a batch of $64 \times 50$ transitions $(o_t, s_t, r_t)$. This batch is then used to compute policy gradients and perform PPO updates. We sample another set of environments for the next batch of rollouts.

**Supervised Fine-Tuning (SFT).** For the SFT experiments, we train the model for 6 epochs on 32k transitions sampled from UCB rollouts in environments drawn from the training task distribution $\mathcal{D}$. Transitions are uniformly sampled across the length of training horizon $T$. We synthesize a templated response for each transition by demonstrating the step-by-step UCB value calculation for each arm and the comparison process which leads to the final action. We perform full fine-tuning minimizing the cross-entropy loss between the predicted and ground-truth responses.

**Bandit Environments.** We primarily consider MAB environments listed in Table 1. Our approaches are also generalizable to a popular contextual bandit setup described in Appendix D. The environments can be generally grouped into two families: Gaussian and Bernoulli, based on the reward distribution. The Gaussian environments have continuous reward distributions, while the Bernoulli environments have discrete binary reward distributions. We select one from each family (i.e., `Bernoulli5_Uniform` and `Gaussian5_Var1_MeanN0`) as the two training task distributions, under which we train two set of policies to test out-of-distribution generalization.

Table 1: Generic families of $k$-armed MAB environments and some specific parameterizations used in our study. Asterisk indicates the training task distributions.

| Family | Reward Dist. | Mean Dist. | Example Instantiation |
|---|---|---|---|
| Gaussian$k$_Var$\sigma^2$_MeanN$m$ | $r \sim \mathcal{N}(u_i, \sigma^2)$ | $u \sim \mathcal{N}(m, \sigma_u^2)$ | Gaussian5_Var1_MeanN0* |
| Gaussian$k$_Var$\sigma^2$_MeanU | $r \sim \mathcal{N}(u_i, \sigma^2)$ | $u \sim \mathcal{U}(0, 1)$ | Gaussian5_Var1_MeanU |
| Bernoulli$k$_Uniform | $r \sim \mathcal{B}(u_i)$ | $u \sim \mathcal{U}(0, 1)$ | Bernoulli5_Uniform* |
| Bernoulli$k$_Delta$\Delta$ | $r \sim \mathcal{B}(u_i)$ | $u_{i^*} = p,$ | Bernoulli5_Delta0.2 |
| | | $u_i = p - \Delta, \forall i \neq i^*$ | |

**Baselines.** We compare learning agents against the following standard baselines: 1) **Upper Confidence Bound (UCB)** (Auer et al., 2002) selects the action $A_t = \arg\max_a \left( Q_t(a) + C \times \sqrt{\frac{\log(t)}{N_t(a)}} \right)$, where $Q_t(a)$ is the mean reward of action $a$ up to time $t$, $N_t(a)$ is the number of times action $a$ has been selected up to time $t$, and $C$ is a constant. 2) **Thompson Sampling (TS)** (Thompson, 1933) is a Bayesian method that samples from the posterior reward distribution of each action and selects the one with the highest sample. We use Beta and Gaussian posteriors for Bernoulli and Gaussian rewards, respectively. 3) $\epsilon$-**Greedy** chooses a random action with probability $\epsilon$ and the action with the highest current mean reward otherwise. While simple, its constant exploration leads to linear regret. The purely exploitative **Greedy** policy is a special case where $\epsilon = 0$ and we perform a round-robin start.

For UCB, which sometimes serves as a teacher, we use an exploration constant of $C = 0.5$, which performs well for both training environments. For $\epsilon$-Greedy, we use a standard $\epsilon = 0.1$; while likely suboptimal, it provides a consistent anchor for comparison. The direct performance comparison between learned agents and baselines is *not* the central focus of this study. The one exception is the evaluation of our imitation learning agents against their UCB teacher.

**Evaluation.** We evaluate the policy over 64 episodes, each with a maximum of 300 steps. We use a fixed set of 64 different seeds for initialization of evaluation environments and baseline policies. To compare the policies and test for length generalization, we follow standard practice to report cumulative regret at $t \in \{50, 300\}$. MAB instances, even when they are drawn from the same distribution, can be quite different in terms of challenge level. Conventional empirical evaluation aggregates from exccessive number of rollouts (e.g., ten of thousands) and long horizons, which although provides a more stable estimate is prohibitively costly for LLM inference. We therefore utilize distribution plots to visualize this variation in regret and focus on the typical performance in comparison. To provide a more comprehensive evaluation, we supplement this with two additional metrics: time-averaged reward and best arm frequency, which measure the proportion of times the optimal arm is selected.

## 5 EXPERIMENTAL RESULTS

### 5.1 LLM AGENTS ARE META-BANDIT LEARNERS

As shown in Figure 3, across both training setups, RL-trained policies improve upon pre-trained models to be comparable with classical baselines (UCB, TS, $\epsilon$-Greedy), achieving lower cumulative regret and length generalization to a $6\times$ longer horizon ($50 \to 300$). The time-averaged reward (AvgReward) and best arm frequency (BestArmFreq) in Table 2 indicate steady performance gains over time. Learning agents remain competitive under OOD evaluation, exhibiting non-trivial cross-distribution transfer from Gaussian-trained policies to Bernoulli environments and vice versa. However, RL agents that trained on environmental feedback (i.e., `RL-OG` and `RL-STG`) show weaker cross-distribution generalization, with greater variability in worst-case performance. Following Nie et al. (2024), we adopt function fitting to measure the linearity versus sub-linearity of cumulative regret through time. We find that pretrained LLM exhibits linear regret (similar to Greedy), while fine-tuning greatly reduces the linearity, reaching levels comparable to or lower to UCB (Appendix B).

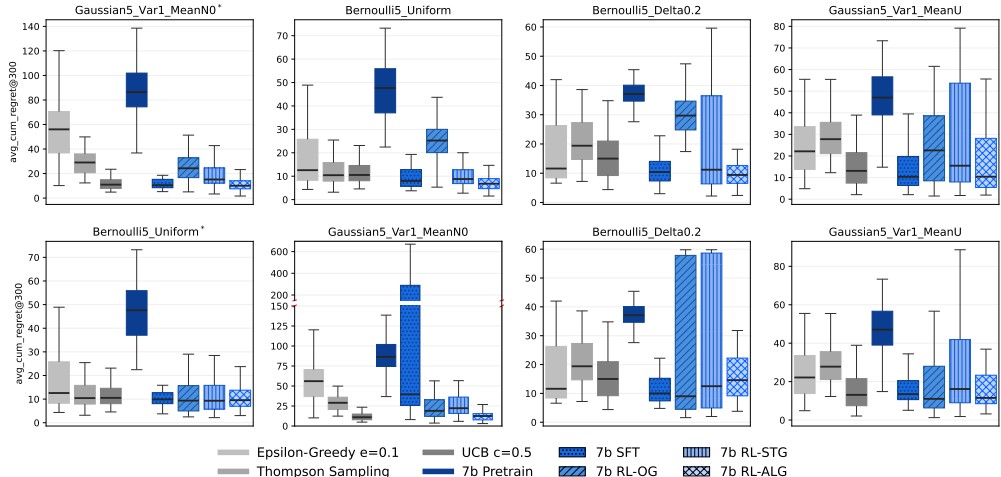

Figure 3: Comparison of LLM policies against baselines on cumulative regret at 300 steps. The first row shows the results of 7B models trained on `Gaussian5_Var1_MeanN0`, and the second row shows the results of 7B models trained on `Bernoulli5_Uniform`. Evaluation is performed both in- (first column) and out-of-distribution (other columns). The boxplots depict the median, interquartile range (IQR) from the $25^{\text{th}}$ to the $75^{\text{th}}$ percentile, and whiskers extending to $1.5\times$IQR. 3B model results can be found in Appendix B.

**Learning from UCB signals.** Overall, policies optimized using teacher UCB signals, whether through reinforcement learning (`RL-ALG`) or supervised fine-tuning (`SFT`), consistently outperform policies trained solely on the task reward signal (`RL-OG`). This underscores the difficulty of training RL LLM policies for long-horizon exploration where credit assignment is challenging. The imitation policies match or achieve lower cumulative regret compared to the teacher UCB policy in all evaluation environments, revealing a seemingly exciting result: policies trained on expert-generated data can ultimately outperform the very expert policy that produced the data.

**Improving training with strategic rewards.** To learn RL policies from environmental feedback, while theoretically aligned with the original bandit reward signal (`RL-OG`), optimizing for strategic rewards (`RL-STG`) empirically improves the performance of the policy in the Gaussian training setup, despite certain instabilities observed in OOD evaluation. As the variance of the reward distribution decreases, `RL-STG` becomes equivalent to `RL-OG`, which explains why their performance is more closely matched when training in the `Bernoulli5_Uniform` environment.

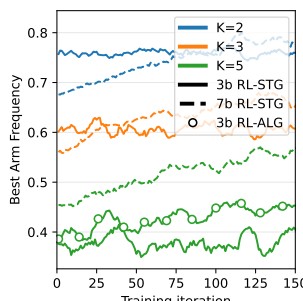

**SFT vs RL for imitation.** SFT with UCB expert demonstrations can achieve similar regret to UCB in-domain, consistent with prior work (Schmied et al., 2025). We additionally find SFT policies to be surprisingly competitive out-of-distribution. Part of the reason is that UCB is a distribution-agnostic policy–the same calculation can be applied to different reward distributions as long as the LLM follows the arithmetic operations. This generalization is however fragile. SFT policies can overfit to the training distribution and cause a degradation of basic arithmetic capability. Together, these factors

Figure 4: Training performance of `RL-STG` policy on `GaussianK_Var1_MeanN0` with 3B and 7B models. We additionally include `RL-ALG` of 3B model (5 arms) for comparison.

lead to higher variability and worst-case regret in OOD evaluation. For example, in Figure 3, the SFT policy trained in the `Bernoulli5_Uniform` environment exhibits unsatisfactory worst-case performance in `Gaussian5_Var1_MeanN0`, while the `RL-ALG` policy based on UCB reward signal remains robust.

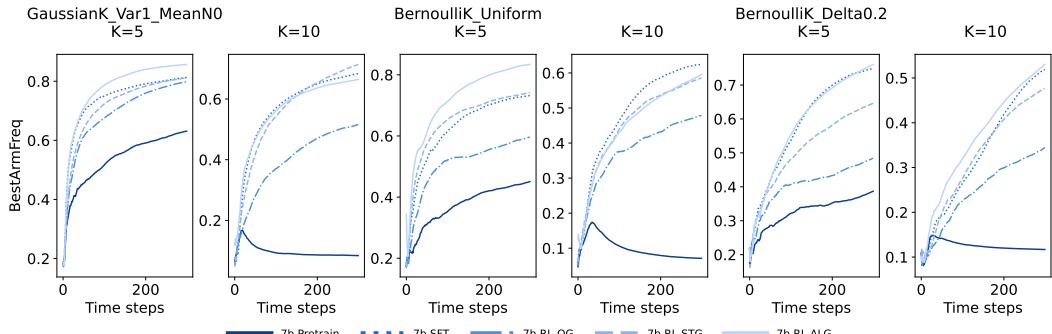

Figure 5: Generalization to environments with 10 arms of 7B LLM policies trained on `Gaussian5_Var1_MeanN0`. Best arm selection frequency is reported.

Table 2: Analytics of baselines and 7B LLM policies trained on `Gaussian5_Var1_MeanN0`, evaluated in-distribution. Except rewards, statistics are shown as percentages. Note that suffix failures emerge from every fine-tuned LLM agents.

| Type | Metric | AvgReward | | BestArmFreq | | GreedyFreq | | SuffixFail | | MinFrac | |
|------|--------|-----------|------|-------------|------|-----------|------|-----------|------|---------|------|
| | @$t$ | 50 | 300 | 50 | 300 | 50 | 300 | 50 | 150 | 50 | 300 |
| *Baselines* | Greedy | 0.91 | 1.01 | 65.4 | 71.7 | 90.0 | 98.3 | 25.0 | 25.0 | 10.2 | 1.7 |
| | $\epsilon$-Greedy | 0.76 | 0.90 | 47.9 | 67.6 | 91.3 | 91.6 | 0.0 | 0.0 | 2.5 | 8.5 |
| | TS | 0.77 | 1.00 | 55.8 | 78.5 | 67.0 | 85.2 | 0.0 | 0.0 | 15.2 | 4.0 |
| | UCB | 0.91 | 1.04 | 67.7 | 80.6 | 83.3 | 95.4 | 3.1 | 4.7 | 10.5 | 1.8 |
| *LLM Agents* | Pretrain | 0.55 | 0.79 | 45.4 | 63.1 | 48.7 | 65.4 | 0.0 | 0.0 | 35.2 | 18.1 |
| | SFT | 0.92 | 1.05 | 69.4 | 81.3 | 83.5 | 95.5 | 6.2 | 6.2 | 10.5 | 1.8 |
| | RL-OG | 0.81 | 1.01 | 61.1 | 79.8 | 78.3 | 91.7 | 1.6 | 4.7 | 13.3 | 3.6 |
| | RL-STG | 0.84 | 1.01 | 63.7 | 81.1 | 83.7 | 95.8 | 3.1 | 6.2 | 10.9 | 2.4 |
| | RL-ALG | 0.92 | 1.05 | 70.7 | 85.7 | 85.4 | 97.0 | 7.8 | 9.4 | 10.0 | 1.7 |

**Small models struggle to learn without teachers.** Figure 4 illustrates the training dynamics of the `RL-STG` policy on the `GaussianK_Var1_MeanN0` environment using 3B and 7B parameter models, measured by the frequency of selecting the best arm. For the 7B models, performance improves over iterations across varying numbers of arms (K=2, 3, 5), with higher accuracy for smaller K. In contrast, the 3B model exhibits stagnant accuracy, starting comparably or even higher than the 7B counterpart in simpler 2- and 3-arm settings pre-training but failing to improve with RL updates. Nevertheless, we observe that the 3B model can learn with teacher guidance using `RL-ALG` or `SFT`. This highlights the challenges of training smaller models with RL on task rewards, as learning effective exploration policies from environmental feedback demands long-horizon credit assignment.

**Generalization to Larger Action Space.** The pre-trained model's performance degrades significantly as the action space increases. In three 10-arm environments, it exhibits excessive regret, while the policies we trained maintain stable performance. Figure 5 illustrates this failure: the base LLM's best-arm selection frequency stops improving at an early stage and degrades to near random performance. All the trained agents do not suffer from this collapse, suggesting that the training process has successfully enhanced the policy's generalization to an increased action space.

We will explore in the following section why the LLM policies excel, what strategies drive their success, and how different learning paradigms lead to different behaviors, to better understand their potential and limitations.

## 5.2 ANALYZING LLM EXPLORATION STRATEGIES

We additionally include three surrogate statistics used in Krishnamurthy et al. (2024) as diagnostics for long-term exploration failure: GreedyFreq@$t$ measures the relative frequency of rounds that

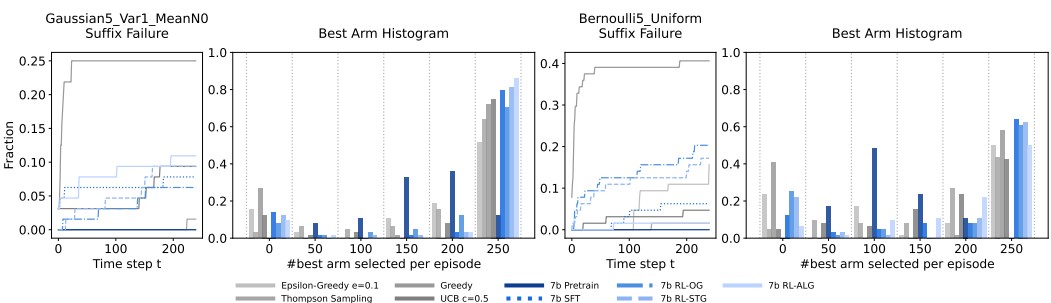

Figure 6: Suffix failure frequency and best arm selection histogram of the two sets of LLM policies described in Figure 3, evaluated in-distribution. We observe that the distributions of best arm selection frequency of learned policies shifts from approximately normal distributions of the pretrained model to bimodal distributions.

selects the greedy action up to time $t$; SuffixFail@$t$ measures the frequency of suffix failures; and MinFrac@$t$ diagnoses uniform-like failures. Specifically, a suffix failure at $t$ indicates that the policy never selects the optimal arm again for rounds $t, \ldots, T$, while MinFrac@$t$ tracks the minimum fraction of rounds any arm is chosen up to $t$, rescaled by $K$ to the range $[0, 1]$. For MinFrac, sustained large values indicate a failure to converge (uniform selection).

**Learned policies exhibit greedy tendencies.** While learned LLM policies achieve lower regret and less uniform failures compared to the pretrained model, our qualitative analysis reveals concerns about suboptimal exploration. The first warning sign, shown in Table 2, is that the learned agents exhibit higher suffix failure frequency than both the pre-trained model and theoretical optimal policies. This indicates premature abandonment of the best arm, a pattern absent in the pretrained model. Learning also alters the distribution of best arm selection frequency from an approximately normal distribution for the pretrained model to a bimodal distribution, where the agent either almost always selects or very rarely selects the best arm within an episode, a characteristic of Greedy behavior (Figure 6). Direct measurement of greedy-arm selection frequency further confirms that the learning agents reach the exploitation phase more quickly than the pre-trained model.

**Dissecting imitation of the UCB oracle: RL vs. SFT.** Although the UCB policy is itself greedy in construction, student policies trained under UCB teacher often amplify this tendency. This partially explains why LLM policies trained to imitate UCB decisions can sometimes perform better than the oracle. To analyze this phenomenon, we compare how often the choices of each UCB-mimicking policy diverge from the oracle's decision given the same state, a metric we refer to as the "match rate". For SFT policies that explicitly calculate confidence-bound values, we additionally report the absolute difference between the policy's predicted UCB value and the corresponding oracle value, averaged across the arms.

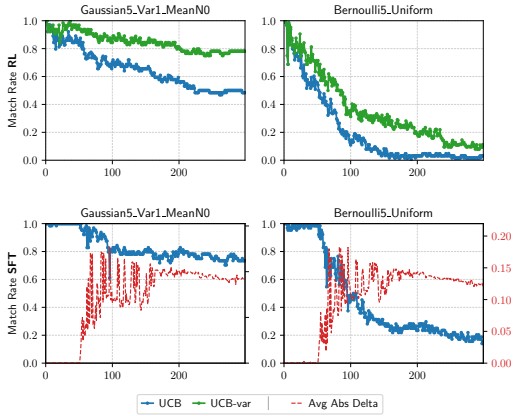

Figure 7: Match rates of the `RL-ALG` policy against decisions of the oracle UCB and a UCB-like algorithm discovered in LLM rationales, at different timesteps. SFT policy shows a jump in calculation errors at the 51st step.

As shown in Figure 7, when both policies trained with UCB supervision in the same environment `Gaussian5_Var1_MeanN0`, the SFT policy maintains a higher match rate than the RL policy from the beginning, indicating that it more faithfully imitates the teacher's decisions. Both sustain a high match rate above 80% in the first 50 steps in-distribution, with the SFT policy tracks the oracle's UCB decisions more closely. However, this stronger imitation capability also makes the SFT policy more susceptible to

overfitting, meaning its high match rates are only guaranteed when evaluation conditions closely resemble the training data.

This sensitivity becomes apparent when training on different data. While the `SFT` policy trained on `Gaussian5_Var1_MeanN0` sustains high match rates across all tested environments, an `SFT` policy trained on `Bernoulli5_Uniform` achieves this *only* within the same Bernoulli family of tasks (Figure 20). We find this failure is a result of systematic errors in simple calculations involving negative rewards, which are unseen during training—a sign of catastrophic forgetting of basic arithmetic skills (Chu et al., 2025; Shenfeld et al., 2025). The agent frequently miscalculates the UCB values and subsequently disregards its own calculations. This leads to asymmetric generalization, where performance degrades sharply outside the training distribution, consistent with the higher worst-case regret we previously observed in Figure 3. These results highlight the critical importance of training data selection to balance imitation fidelity with robustness.

**What incentivizes RL-ALG to prioritize exploitation over imitation?** The adaptive behavior of the `RL-ALG` policy is a subtle consequence of the bandit learning structure and a fundamental change in the UCB teacher's behavior over an episode. Initially, the UCB algorithm balances high uncertainty (exploration) and high observed rewards (exploitation). As an episode progresses, the uncertainty bounds shrink, and the teacher's policy converges. Its decisions become increasingly dominated by the empirical means, causing it to select the greedy arm. In this regime, the RL objective, though formally defined in terms of imitation, becomes highly correlated with a reward for exploitation. The agent discovers that it can optimize more easily by directly picking the greedy arm, rather than faithfully internalizing the teacher's complex exploration logic.

**LLM rationales reveal flawed, exploitative heuristics.** The LLM's generated rationales can reveal its underlying decision-making process. RL policies trained on bandit rewards converge to templated heuristics that most oftenly compare and choose the arm with the highest mean reward. Their explorative actions are driven by rationales that explicitly evaluate the uncertainty of the arms, sometimes with UCB-like calculation. `RL-ALG` trained in `Gaussian5_Var1_MeanN0` however converges to a UCB-like algorithm and mentions in its rationales 98% of the time: $Q_t(a) + C \times \sqrt{\frac{\log(N_t(a)+1)}{N_t(a)}}$. In the standard UCB algorithm, the numerator of the exploration term $\log(t)$ grows with the total number of pulls, ensuring that no action is ever abandoned permanently. In contrast, this learned variant's exploration term depends only on $N_t(a)$, the pulls of a specific arm. This allows the policy to prematurely stop exploring an action if it appears unprofitable in the short run, embodied an exploitative tendency.

Figure 7 shows that this UCB-like algorithm describes the LLM policy better than the oracle UCB. However, the LLM does not follow the algorithm strictly, as its decisions are also affected by frequent numerical inaccuracies such as miscalculating the $\log$ term. We can also observe that when the policy diverges from the UCB variant's decisions, it opts for the greedy action more than 86% of the time. In `Bernoulli5_Uniform`, the LLM converges to another variant $Q_t(a) + \frac{C}{\sqrt{N_t(a)}}$ with similar greedy behavior. These findings reveal that the `RL-ALG` policy learns approximate, error-tolerant variants of UCB, blending imitation with opportunistic exploitation that lowers regret in certain environments. Intriguingly, we observe that the LLM generates the correct UCB formula with inaccurate calculations during early training stages. Its eventual convergence to greedy variants suggests failures in credit assignment.

## 6 Conclusion

We fine-tune LLM agents via SFT and RL with novel reward signals, achieving strong performance with lower regret and robust generalization to $6\times$ longer horizons and new reward distributions in the multi-armed bandit task. However, behavioral analysis reveals that training elicits short-sighted, exploitative policies. This emergent greediness is a consequence of the fundamental imbalance in training data, where sparse exploration signals are easily overwhelmed by frequent exploitation. Compounded by the complex credit assignment problem, this challenge highlights the need for methods that explicitly amplify exploration signals. Future work could explore focused replay techniques that re-weight experiences based on information gain and surprise or design adversarial and curriculum-based environments that make robust long-horizon planning a necessity for success.

## REPRODUCIBILITY

To support the reproducibility of our results, we provide more implementation details in Appendix A. We commit to sharing the code, pre-trained models and data to the general public upon publication.

## ACKNOWLEDGMENTS

We thank Ronald Parr, Shuyan Zhou, and Fan Yao for providing insightful feedback and suggestions to the manuscript. We also thank Qian Liu, Weiting Tan, and Junlin Wang for helpful discussions. This work was supported by NSF award IIS-2211526.

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

# A    IMPLEMENTATION DETAILS

We report additional details for the environment settings, the RL and SFT training of LLM policies.

## A.1    ENVIRONMENT SETTINGS

Table 3: Generic families of $k$-armed MAB environments and a complete list of 15 parameterizations used in our study. Asterisk indicates the training task distributions.

| Family | Reward Dist. | Mean Dist. | Example Instantiation |
|---|---|---|---|
| Gaussian$k$_Var$\sigma^2$_MeanN$m$ | $r \sim \mathcal{N}(u_i, \sigma^2)$ | $u \sim \mathcal{N}(m, \sigma_u^2)$ | Gaussian5_Var1_MeanN0* |
| | | | Gaussian10_Var1_MeanN0 |
| | | | Gaussian5_Var3_MeanN0 |
| | | | Gaussian5_Var1_MeanN±1 |
| | | | Gaussian5_Var3_MeanN±1 |
| Gaussian$k$_Var$\sigma^2$_MeanU | $r \sim \mathcal{N}(u_i, \sigma^2)$ | $u \sim \mathcal{U}(0, 1)$ | Gaussian5_Var1_MeanU |
| | | | Gaussian5_Var3_MeanU |
| | | | Gaussian5_Var5_MeanU |
| Bernoulli$k$_Uniform | $r \sim \mathcal{B}(u_i)$ | $u \sim \mathcal{U}(0, 1)$ | Bernoulli5_Uniform* |
| | | | Bernoulli10_Uniform |
| Bernoulli$k$_Delta$\Delta$ | $r \sim \mathcal{B}(u_i)$ | $u_{i^*} = p,$ $u_i = p - \Delta, \forall i \neq i^*$ | Bernoulli5_Delta0.2 |
| | | | Bernoulli10_Delta0.2 |
| | | | Bernoulli5_Delta0.1 |

We evaluate our policies on a comprehensive set of environments from the Gaussian and Bernoulli families, as listed in Table 3. The `Bernoullik_Delta`$\Delta$ class, studied by Krishnamurthy et al. (2024), allows for instance difficulty to be easily adjusted by changing the reward gap between the optimal and sub-optimal arms. However, this environment is unsuitable for training, as a policy can simply explore each action sequentially until finding one with a mean reward above 0.5. The `Gaussiank_Var`$\sigma^2$`_MeanN`$m$ class is a popular benchmark introduced in Sutton et al. (1998) and later used by Nie et al. (2024). In these environments, the variances of both the reward and mean distributions are tied to a single hyperparameter, $\sigma^2$. Increasing the variance does not necessarily make the environment more challenging; while the rewards become noisier, the means also become more dispersed. These two effects offset each other, so this comparison group primarily tests a policy's robustness to multiplicative rescaling and shifts in rewards. The `Gaussiank_Var`$\sigma^2$`_MeanU` class, from Schmied et al. (2025), maintains a fixed uniform distribution for the means, allowing the reward variance to be adjusted via $\sigma^2$ to control for different difficulty levels.

The diverse distribution shifts across these test environments—including changes in mean, variance, and distributional family—are designed to ablate different aspects of generalization. For instance, transitioning from `Gaussian5_Var1_MeanN0` to `Gaussian5_Var1_MeanN±1` provides a targeted test of a policy's ability to handle shifted reward distributions while all other properties remain constant. To assess scalability, we additionally test the policies on environments with an increasing number of arms (from $k = 5$ to $k = 10$).

## A.2    RL SETTINGS

Table 4 presents the hyperparameters used for PPO training of the LLM policies (`RL-OG` and `RL-STG`). For the algorithmic-reward variant (`RL-ALG`), we retain all settings except that we set the episode-level discount factor and GAE lambda to zero, since cumulative rewards are not required.

We use vLLM (Kwon et al., 2023) for asynchronous rollouts across parallel environments and FSDP (Zhao et al., 2023) for fully sharded training under the VeRL framework. At the start of each iteration, all environments are reinitialized with fresh random seeds to ensure diverse experience collection.

Table 4: Hyperparameters for the PPO training of LLM policies (`RL-OG` and `RL-STG`).

| Category | Hyperparameter | Value |
|---|---|---|
| *Model & Environment* | | |
| Model | Base Language Model | Qwen/Qwen2.5-3/7b-Instruct |
| | Max Response Length | 1024 tokens |
| | Temperature | 1.0 |
| Environment | Type | Various (Gaussian, Bernoulli) |
| | Number of Arms ($k$) | 5 |
| | Episode Length ($T$) | 50 |
| | Number of Parallel Environments | 64 |
| *PPO Algorithm* | | |
| Optimization | Optimizer | AdamW (Kingma & Ba, 2015) |
| | Actor Learning Rate ($\alpha_\pi$) | $1 \times 10^{-6}$ |
| | Critic Learning Rate ($\alpha_V$) | $1 \times 10^{-5}$ |
| | Gradient Clipping | 1.00 |
| | Response-level Discount Factor ($\gamma_{\text{intra}}$) | 1.00 |
| | Response-level GAE Lambda ($\lambda_{\text{intra}}$) | 1.00 |
| | Episode-level Discount Factor ($\gamma_{\text{inter}}$) | 0.95 |
| | Episode-level GAE Lambda ($\lambda_{\text{inter}}$) | 0.95 |
| | PPO Clipping Coefficient ($\epsilon$) | 0.20 |
| | PPO Mini-batch Size | 128 |
| Regularization | Weight Decay | $1 \times 10^{-2}$ |
| | Entropy Coefficient | $5 \times 10^{-4}$ |
| *Training Infrastructure* | | |
| Training | Total Training Steps | 500 |
| Hardware | Number of GPUs | 4 |
| | Tensor Parallelism (Rollout) | 4 |

Model checkpoints are saved every 100 training steps. Each checkpoint is evaluated on the same set of environments (matching the training type) to guarantee fair comparison. The checkpoint with the lowest cumulative regret is selected as the final model.

### A.3 SFT SETTINGS

We generate training data for supervised fine-tuning by sampling $N$ trajectories from the environment using a UCB policy. To expose the model to a broad spectrum of environment configurations and exploration behaviors, we uniformly sample states and actions across each trajectory's horizon.

As in our reinforcement learning experiments, we save model checkpoints at the end of every epoch and evaluate them on the same set of environments to ensure a fair comparison.

Table 5: Hyperparameters for Supervised Fine-Tuning (SFT).

| Category | Hyperparameter | Value |
|---|---|---|
| | *Model & Data* | |
| Model | Base Language Model | Qwen/Qwen2.5-3/7b-Instruct |
| Data | Type | Various (Gaussian, Bernoulli) |
| | Number of Arms ($k$) | 5 |
| | Max Episode Length ($T$) | 50 |
| | Number of Examples ($N$) | 32768 |
| | *Optimization* | |
| Optimizer | Type | AdamW |
| | Learning Rate | $1 \times 10^{-5}$ |
| | Betas ($\beta_1, \beta_2$) | (0.9, 0.95) |
| | Weight Decay | 0.01 |
| | LR Scheduler | Cosine Decay |
| | Warmup Ratio | 0.1 |
| | Gradient Clipping | 1.0 |
| | *Training Details* | |
| | Batch Size | 256 |
| | Epochs | 6 |

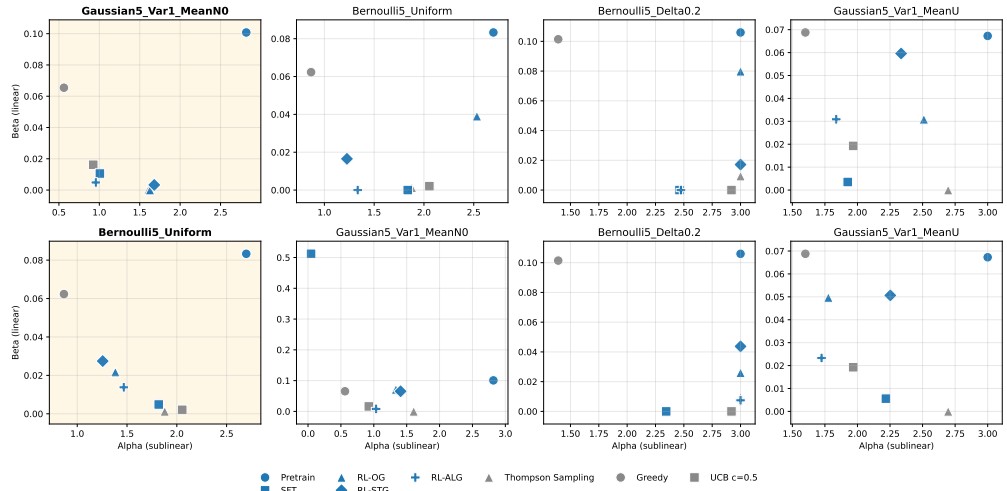

Figure 8: Comparison of LLM policies against baselines on cumulative regret function fitting results. The first row shows the results of 7B models trained on `Gaussian5_Var1_MeanN0`, and the second row shows the results of 7B models trained on `Bernoulli5_Uniform`. Evaluation is performed both in- (first column, highlighted) and out-of-distribution (other columns) as in Figure 3.

## B  DETAILS OF PERFORMANCE COMPARISON

We provide comprehensive experimental results of the LLM policies compared against baselines over a range of environments and model sizes on cumulative regret at 50 and 300 steps. The observation is consistent with the analysis in main text.

### B.1  7B MODEL COMPARISONS

The cumulative regret trends are consistent across evaluations at 50 and 300 steps. However, the longer 300-step horizon more prominently exposes the weaknesses of simple heuristics like $\epsilon$-greedy. At 50 steps, the imitation learning policies (`RL-ALG` and `SFT`) show some instability, likely due to the stronger mimicking effect of the UCB oracle in the initial exploratory phase.

The learned policies demonstrate effective generalization to higher variance in the `Gaussian5_Var3_MeanN0` environment, where the difficulty of identifying the optimal arm is comparable to the `Gaussian5_Var1_MeanN0` training setting. This generalization fails, however, when increased variance makes the task harder, as is the case in the `Gaussian5_Var`$\sigma^2$`_MeanU` environments. Here, exploration strategies developed in low-variance settings are hindered by their greedy bias, leading to poor performance under high uncertainty. Amidst this, the `RL-OG` policy begins to show a slight advantage. Finally, we note that the `SFT` policy, when trained on `Bernoulli5_Uniform`, consistently fails to generalize to any Gaussian environment (Figure 12).

To quantify the asymptotic behavior of the cumulative regret within 300 steps, we follow Nie et al. (2024) to employ a function fitting approach. We model the cumulative regret through time as $y(t) = \lambda_1 \log(1+t)^\alpha + \beta t$, where $\beta$ captures linear failure modes and the logarithmic term captures exploration costs. In Figure 8, we find that pretrained LLMs exhibit a high linear coefficient ($\beta$), comparable to the Greedy baselines, indicating a failure to converge to the optimal arm. Learned policies instead show a significant reduction in $\beta$ across both in-distribution and out-of-distribution evaluations, suggesting that fine-tuning minimizes asymptotic linear regret in-average. The consistently low MSE values and the visual alignment of the fitted curves validate that this parametric model accurately captures the regret dynamics of the evaluated agents (examples in Figure 9). **We note that**, similar to other regret-based analysis, this function-fitting approach is less sensitive to early failure signals than, for example, suffix fail rate. Since cumulative regret is averaged across episodes, early failures can be disguised by early wins resulting from over-exploitation.

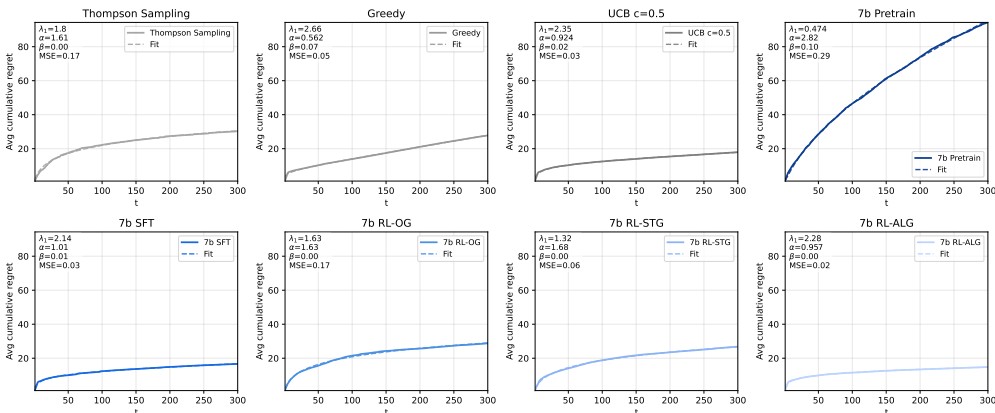

Figure 9: Comparison of LLM policies against baselines on cumulative regret function fitting results. The second row shows the results of 7B models trained on Gaussian5_Var1_MeanN0. Evaluation is performed in-distribution.

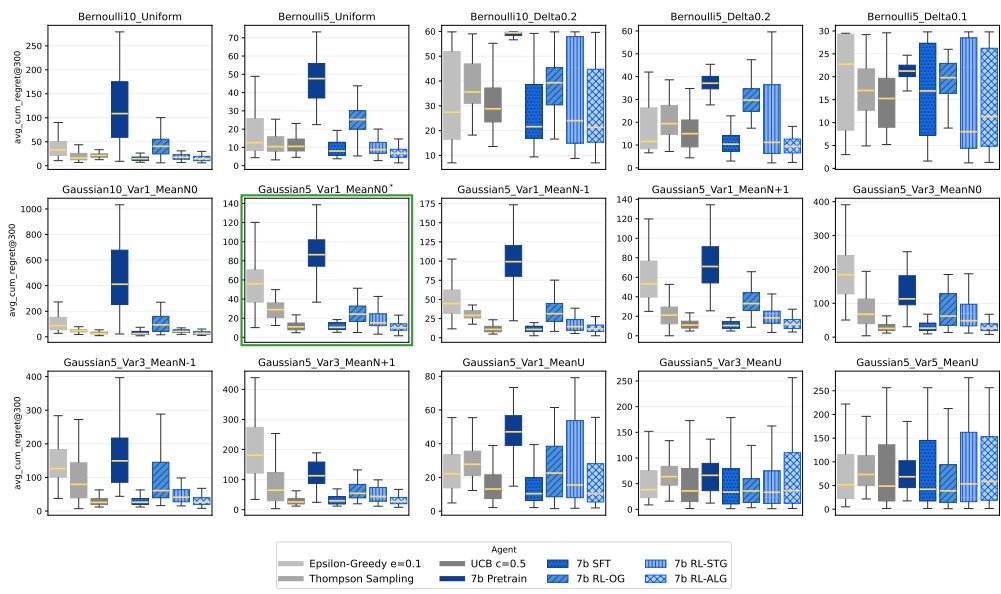

Figure 10: Comparison of LLM policies (7B base model) against baselines on cumulative regret at **300 steps** (outliers are trimmed). Results on training environment has a colored border.

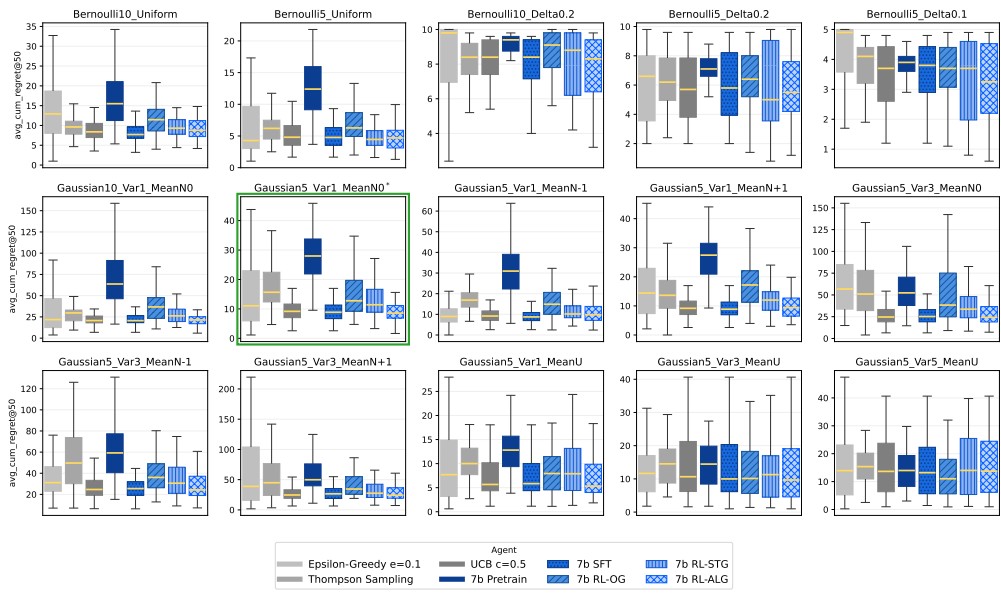

Figure 11: Comparison of LLM policies (7B base model) against baselines on cumulative regret at **50 steps** (outliers are trimmed). Results on training environment has a colored border.

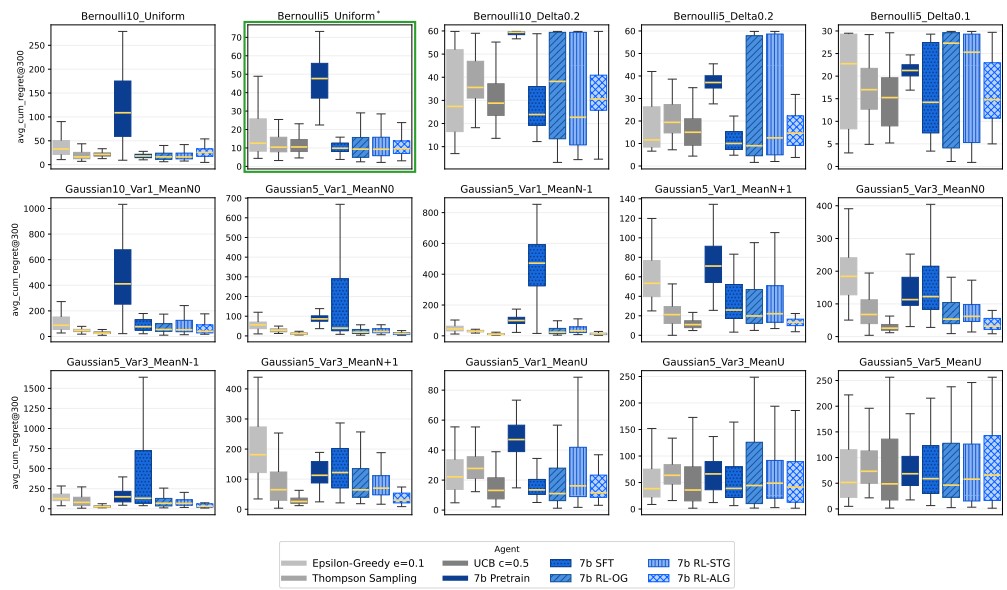

Figure 12: Comparison of LLM policies (7B base model) against baselines on cumulative regret at **300 steps** (outliers are trimmed). Results on training environment has a colored border.

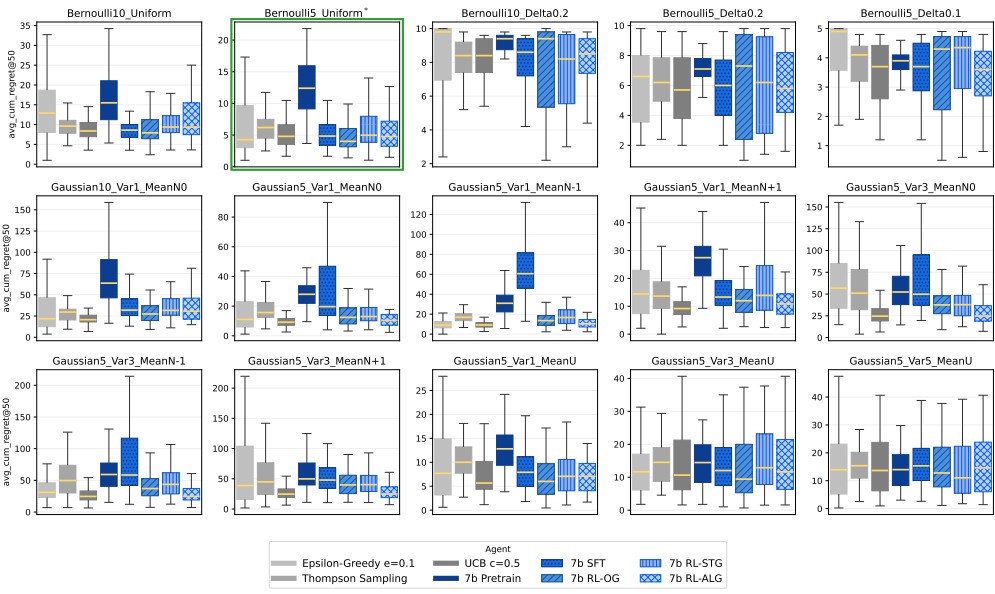

Figure 13: Comparison of LLM policies (7B base model) against baselines on cumulative regret at **50 steps** (outliers are trimmed). Results on training environment has a colored border.

## B.2  3B MODEL COMPARISONS

Table 6: Analytics of baselines and 3B LLM policies trained on `Gaussian5_Var1_MeanN0`, evaluated in-distribution.

| Metric | AvgReward | | BestArmFreq | | GreedyFreq | | SuffixFail | |
|---|---|---|---|---|---|---|---|---|
| @$t$ | 50 | 300 | 50 | 300 | 50 | 300 | 50 | 150 |
| *Baselines* | | | | | | | | |
| Epsilon-Greedy e=0.1 | 0.76 | 0.90 | 47.9 | 67.6 | 91.3 | 91.6 | 0.0 | 0.0 |
| Thompson Sampling | 0.77 | 1.00 | 55.8 | 78.5 | 67.0 | 85.2 | 0.0 | 0.0 |
| Greedy | 0.91 | 1.01 | 65.4 | 71.7 | 90.0 | 98.3 | 25.0 | 25.0 |
| UCB c=0.5 | 0.91 | 1.04 | 67.7 | 80.6 | 83.3 | 95.4 | 3.1 | 4.7 |
| *Learned Agents* | | | | | | | | |
| 3b Pretrain | 0.39 | 0.43 | 29.3 | 32.7 | 85.9 | 67.8 | 12.5 | 12.5 |
| 3b SFT | 0.92 | 1.04 | 69.7 | 80.8 | 83.5 | 93.9 | 6.2 | 9.4 |
| 3b RL-OG | 0.41 | 0.69 | 31.5 | 49.1 | 81.0 | 76.0 | 6.2 | 7.8 |
| 3b RL-STG | 0.51 | 0.75 | 39.3 | 56.9 | 77.7 | 71.5 | 0.0 | 1.6 |
| 3b RL-ALG | 0.81 | 0.95 | 60.2 | 71.2 | 81.7 | 89.0 | 14.1 | 17.2 |

Consistent with our main findings, results in Figure 15 and Figure 17 confirm that smaller models benefit less from reinforcement learning optimized directly for environmental reward signals. The `RL-OG` and `RL-STG` policies trained this way perform on par with the pre-trained model at 50 steps and achieve only small gains at 300 steps with `RL-STG` generally outperforming `RL-OG`.

In contrast, both the imitation learning policies, `RL-ALG` and `SFT`, demonstrate a significant improvement over the pre-trained model. The `SFT` policy, in particular, emerges as the top-performing method, achieving reliably lower regret across nearly all environments. This suggests that, even in the imitation learning setting, smaller models struggle with reinforcement learning optimization process itself. According to Table 6, `SFT` achieves AvgReward and BestArmFreq comparable to the UCB teacher in-distribution. Its performance in identifying the best arm continues to improve with more trials, even in most out-of-distribution environments (Figure 14). We once again observe the previously noted generalization failure of `SFT` at this model size, where it fails to transfer from the `Bernoulli5_Uniform` environment to Gaussian environments with negative rewards (Figure 17).

The pre-trained Qwen 2.5 3B model exhibits a distinct exploration pattern compared to its 7B counterpart. While the pre-trained 7B model starts an episode with high exploration and becomes more exploitative over time, the 3B model begins with a highly greedy strategy (GreedyFreq $\approx 86\%$) and becomes more explorative. This causes its ability to identify the best arm plateaus very early in the `Gaussian5_Var1_MeanN0` environment. These behavioral differences lead to different training dynamics: the 7B model consistently reduces exploration throughout the training iterations, while the 3B model first undergoes a phase of increasing exploration before reducing it until convergence.

Across the board, all learned agents show lower GreedyFreq at 50 steps than the pre-trained model. As trials progress, the two successful imitation learning policies (`RL-ALG` and `SFT`) adopt a more greedy exploitation strategy. As a result, they both suffer from a higher suffix failure rate compared to RL policies trained on environmental feedback. This reinforces our conclusion that their performance gains are associated with more sophisticated greedy policies.

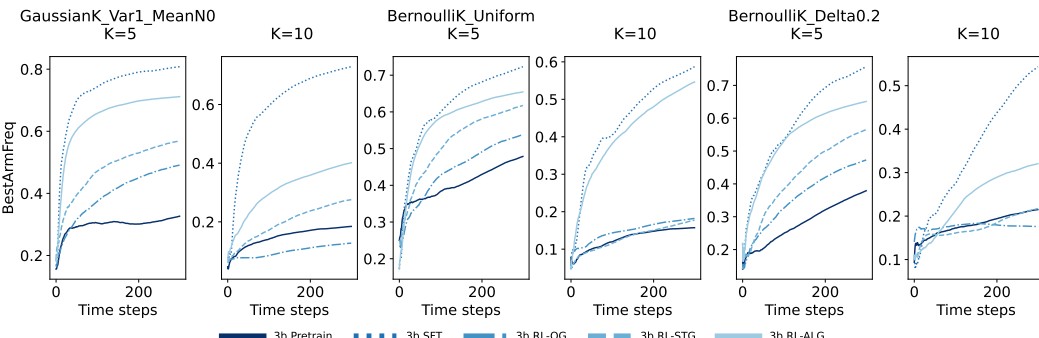

Figure 14: Generalization to environments with 10 arms for Gaussian-trained LLM policies (3B base model). Best arm selection frequency is reported.

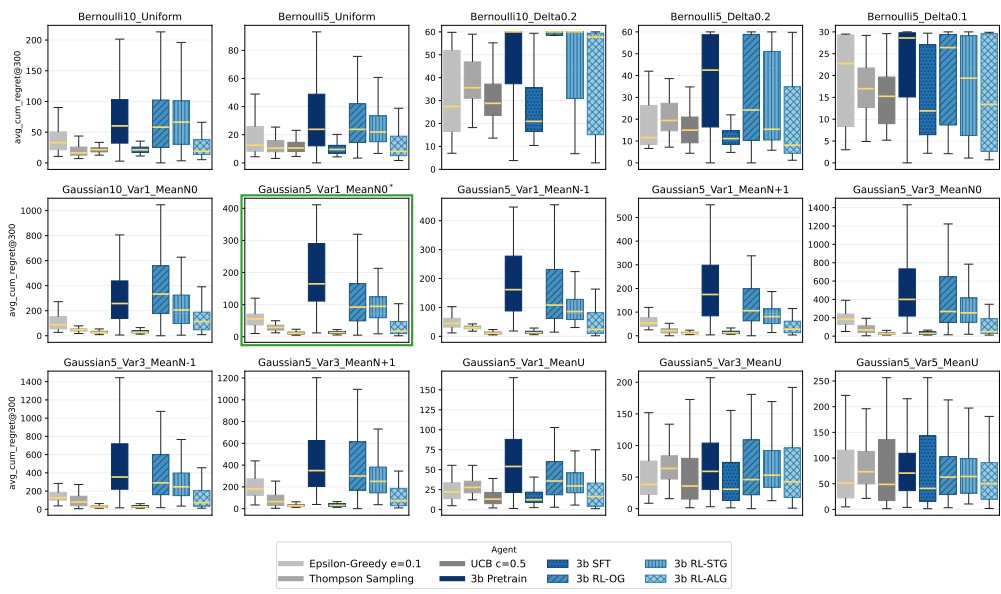

Figure 15: Comparison of LLM policies (3B base model) against baselines on cumulative regret at **300 steps** (outliers are trimmed). Results on training environment has a colored border.

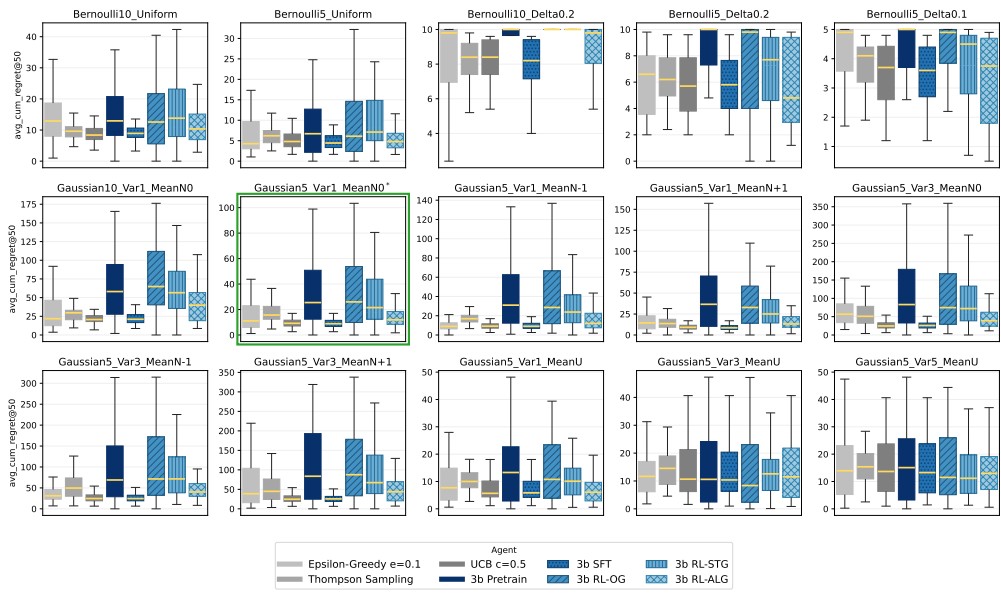

Figure 16: Comparison of LLM policies (3B base model) against baselines on cumulative regret at **50 steps** (outliers are trimmed). Results on training environment has a colored border.

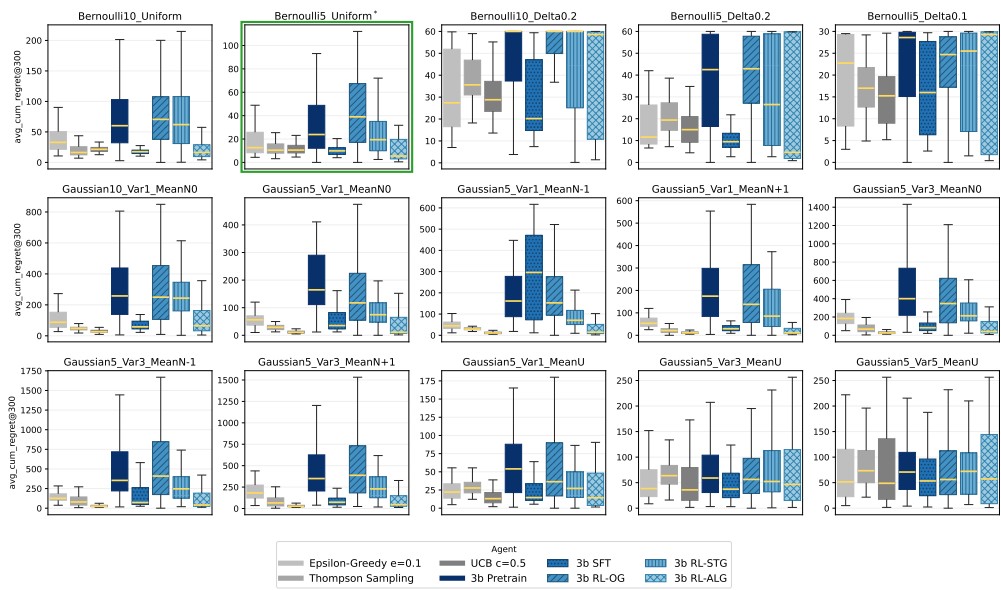

Figure 17: Comparison of LLM policies (3B base model) against baselines on cumulative regret at **300 steps** (outliers are trimmed). Results on training environment has a colored border.

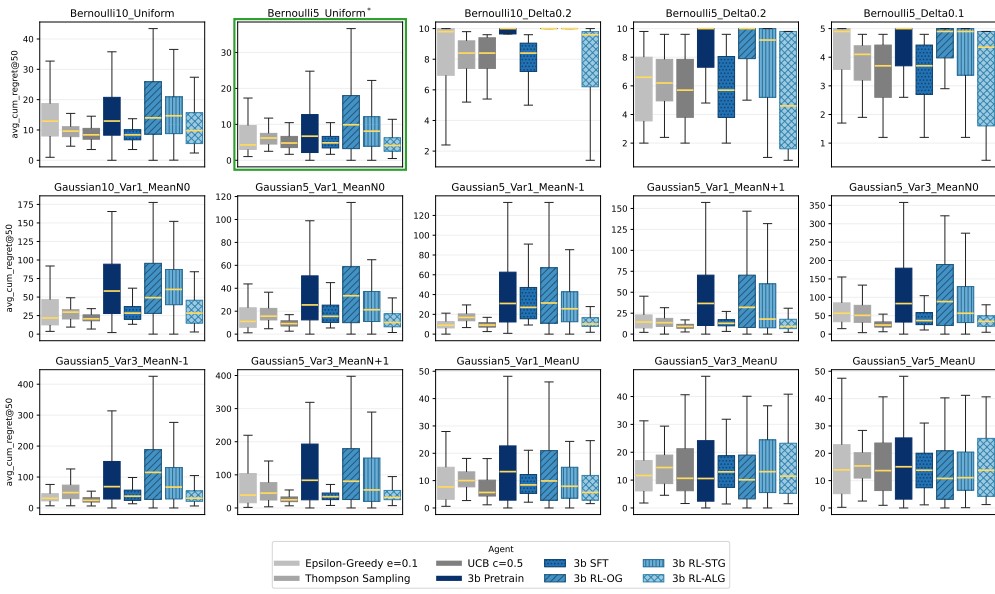

Figure 18: Comparison of LLM policies (3B base model) against baselines on cumulative regret at **50 steps** (outliers are trimmed). Results on training environment has a colored border.

## C    DETAILS OF IMITATION LEARNING ANALYSIS

We investigate why an imitation learning policy might outperform its teacher by analyzing its adherence to key decision-making heuristics. This section expands upon the main text by presenting results from a complete set of experimental environments.

A key finding is that both imitation learning policies (`RL-ALG` and `SFT`) make fewer (imitation) errors in high-variance environments. This is attributed to the teacher UCB policy (C=0.5) itself behaving more greedily in these settings, matching the exploitative bias of the imitation learning policies.

We find that the SFT agent's mistakes reveal errors in both simple arithmetic (summation, subtraction) and complex calculations (logarithms, square roots). A prominent failure mode emerges when the Bernoulli-trained policy observes negative rewards: it often struggles with summations involving these numbers and subsequently disregards its own UCB calculations. For instance, in the `Gaussian5_Var3_MeanN0` environment, the agent chooses an arm different from the one with the highest calculated UCB value 78% of the time. This divergence is sensitive to the reward distribution; lowering the environment's mean reward by 1 increases this deviation rate to 89%, while raising the mean by 1 reduces it to 44%. This behavior indicates a regression in the LLM's capabilities, leading to hallucinations in its reasoning. Future work can explore mixed training with mathematical data to alleviate this issue.

We previously discovered that the `RL-ALG` agents converge to suboptimal variants of the UCB algorithm. This finding is both interesting and disappointing. On one hand, it demonstrates that agents can discover novel solutions from sparse reward signals received only at the end of a response. On the other hand, it suggests that either the oracle policy is not encountered during RL exploration or that credit assignment is a significant challenge. By manually inspecting rollouts from early training iterations, we find that the correct UCB formula did appear, but its calculations were frequently incorrect due to the base model's weakness in complex operations like square roots and logarithms (Figure 23). This points to a credit assignment issue, where the agent incorrectly attributes poor outcomes to the formula itself, rather than to flawed calculations or suboptimal hyper-parameter choices. Future work could explore more fine-grained RL signals to address this problem.

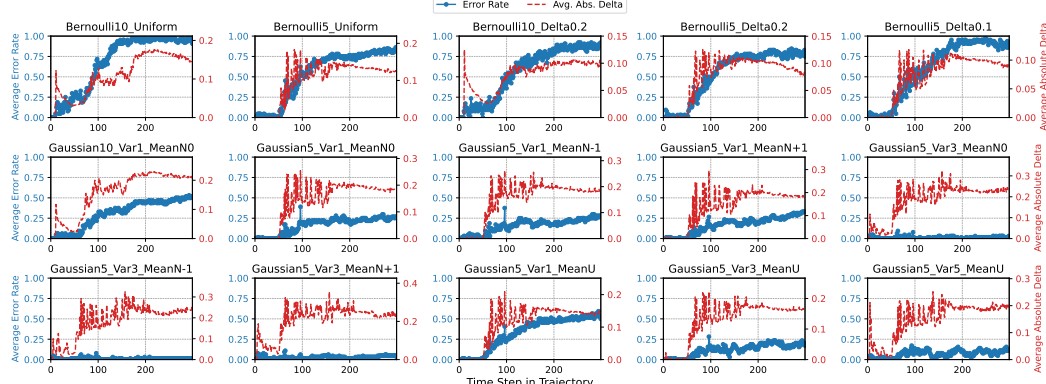

Figure 19: 7B SFT agent trained on Gaussian environments: UCB error by step.

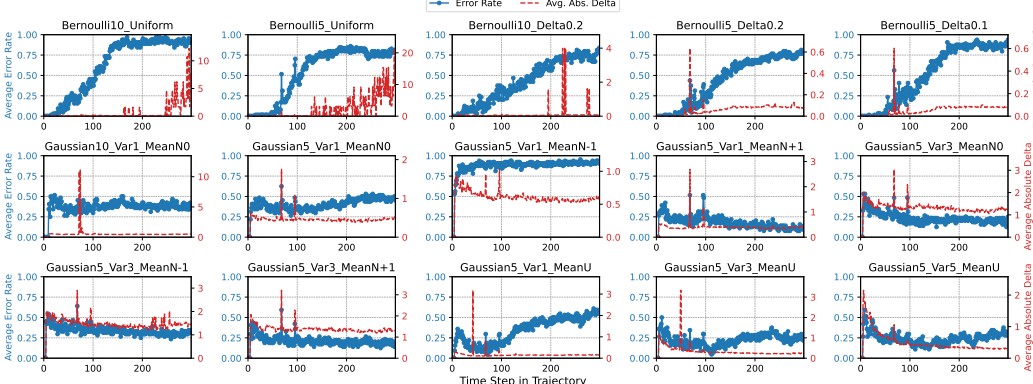

Figure 20: 7B SFT agent trained on Bernoulli environments: UCB error by step.

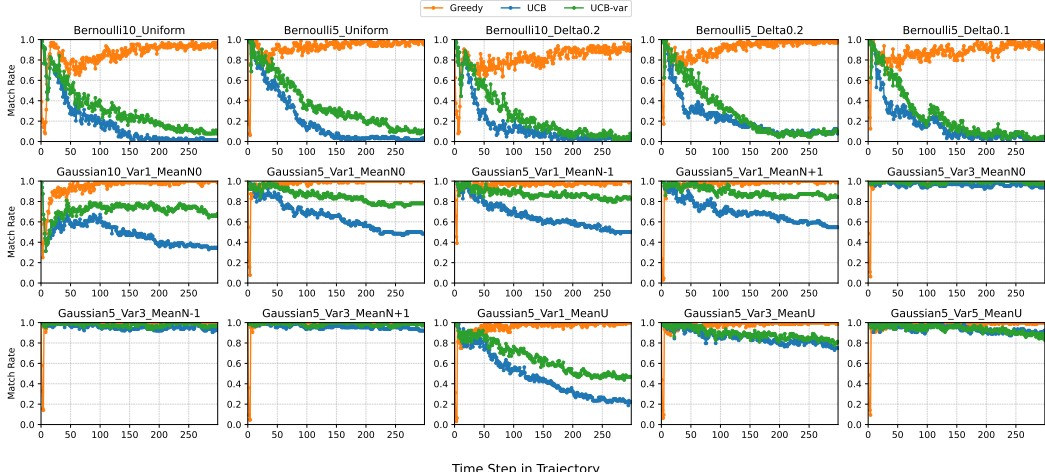

Figure 21: 7B `RL-ALG` agent trained on Gaussian environments to optimize UCB reward signal: match rate by step. UCB_Var here is the UCB variant $Q_t(a) + C \times \sqrt{\frac{\log(N_t(a)+1)}{N_t(a)}}$, which the agent discovered and consistently used.

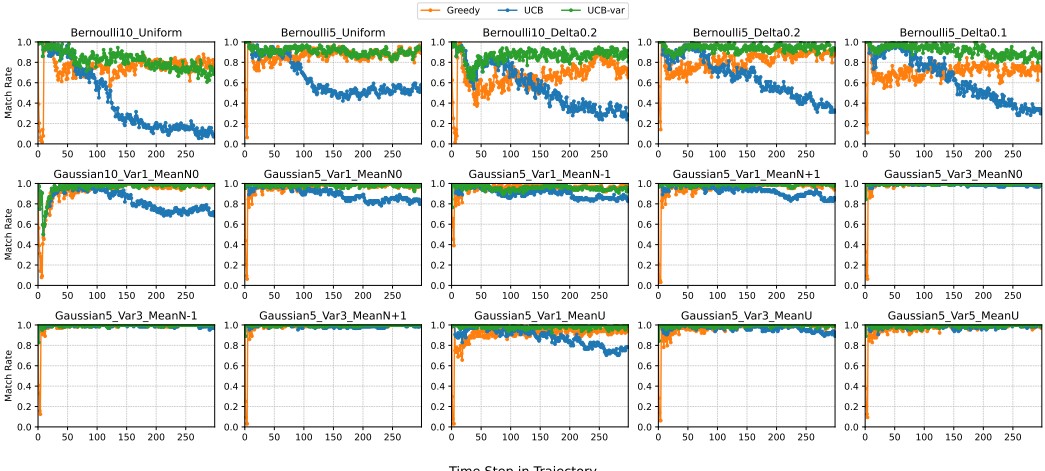

Figure 22: 7B `RL-ALG` agent trained on Bernoulli environments to optimize UCB reward signal: match rate by step. UCB_Var here is the UCB variant $Q_t(a) + \frac{C}{\sqrt{N_t(a)}}$ that the agent discovered and consistently used.

# D    Contextual Bandits on MovieLens

To demonstrate the generalization capabilities of our RL approaches beyond Multi-Armed Bandits (MAB), we conducted experiments on a Contextual Bandit setting using the MovieLens environment from BanditBench (Nie et al., 2024).

## D.1    Environment Setup

Using the open-source implementation of BanditBench, we construct a contextual bandit task based on the MovieLens-100K dataset (Harper & Konstan, 2015). The dataset contains 100,000 ratings for 1,682 movies from 943 users, including demographic information such as age, gender, and occupation. Only the first 100 users are seen during training, the rest of them are used for evaluation.

To align with the standard linear assumption used in baselines like LinUCB, the environment is constructed via low-rank approximation. Specifically, the top $K$ popular movies are selected to form the action space. A user preference matrix $P \in \mathbb{R}^{N \times K}$ is constructed and decomposed using Singular Value Decomposition (SVD) such that $P \approx U\Sigma V^T$. Here, $U$ represents the user embedding matrix, $V$ the movie embedding matrix, and $\Sigma$ the diagonal matrix of singular values. The ground-truth reward for a user $i$ and movie $j$ is deterministic: $r_{i,j} = u_i^T \Sigma v_j$.

## D.2    Experimental Results

Following the same training setting described in the MAB task, we fine-tune the Qwen 2.5 7B model over a horizon of 50 steps with 5 arms (recommending from 5 top movies). Note that as the environment has deterministic reward, RL-STG is equivalent to RL-OG. We compare our methods against the standard Pretrained LLM and LinUCB (Chu et al., 2011) (exploration ratio 0.5).

As shown in Table 7, both RL with environmental feedback (RL-OG) and RL with algorithmic reward (RL-ALG) significantly reduce regret compared to the Pretrained LLM. Notably, our multi-turn RL implementation (RL-OG) outperforms the strong LinUCB baseline. We observe that RL-ALG performs worse than RL-OG; this is likely because RL-ALG attempts to approximate the exact LinUCB calculation—a task more complex than standard UCB—often resulting in a greedy approximation where the model converges to the most frequently chosen action of the teacher.

Table 7: Cumulative Regret on MovieLens (Horizon=50, 5 Arms). Results show mean and standard error across 64 episodes varying random seeds.

| Method | Cumulative Regret |
| --- | :---: |
| LinUCB | 61.09 ($\pm$2.57) |
| 7B Pretrain | 100.84 ($\pm$3.09) |
| 7B RL-OG (full history) | **46.64** ($\pm$1.70) |
| 7B RL-OG (no history) | 58.71 ($\pm$1.32) |
| 7B RL-ALG | 73.23 ($\pm$1.78) |

## D.3    Analysis and Limitations

While these results appear promising, a fundamental limitation exists in using this specific MovieLens setup for meta-bandit training. Unlike MAB settings where we can generate infinite environments with varying parameters, this MovieLens contextual bandit setup represents a single bandit instance defined by a static dataset.

Because the action set is fixed and the linear function defined by $\Sigma$ is universal across all interactions, the task essentially requires the model to map user features (provided in the prompt) to a reward distribution via a static function. This allows the model to *memorize* the underlying function during training rather than performing online exploration. This hypothesis is supported by our ablation study in Table 7, where the **RL-OG (no history)** agent, which has no access to previous interaction feedback, still achieves lower regret (58.71) than the LinUCB baseline (61.09). This indicates the model is relying more on memorized patterns than in-context bandit learning. Consequently, while

our approach generalizes to this setting, we prioritize controllable MAB benchmarks in the main text to ensure rigorous evaluation of the model's online learning and exploration capabilities. We leave the meta-bandit setup of realistic contextual bandit scenarios to future work.

---

**MovieLens Prompt Example**

You are an AI movie recommendation assistant for a streaming platform powered by a bandit algorithm that offers a wide variety of films. There are 5 unique movies you can recommend, named
1. Star Wars (1977)
2. Contact (1997)
3. Fargo (1996)
4. Return of the Jedi (1983)
5. Liar Liar (1997)

When a user visits the streaming platform, you assess their demographic description to choose a movie to suggest. You aim to match the user with movies they are most likely to watch and enjoy. Each time a user watches a recommended movie, you adjust your recommendation algorithms to better predict and meet future user preferences.

So far you have interacted 5 times with the most recent following choices and rewards:

Context: This person is a 39-year-old man, working as a scientist and live in University park of Cook county, Illinois. User preference vector: [-0.01, 0.04, -0.01, -0.04, 0.00].
Recommended movie 2
Reward: 4

Context: This person is a 29-year-old woman, working as a administrator and live in Crofton of Anne Arundel county, Maryland. User preference vector: [-0.05, 0.04, 0.05, 0.01, 0.01].
Recommended movie 1
Reward: 5

Context: This person is a 36-year-old woman, working as a writer and live in Los angeles of Los Angeles county, California. User preference vector: [-0.02, -0.01, -0.07, 0.05, 0.01].
Recommended movie 1
Reward: 0

Context: This person is a 45-year-old woman, working as a educator and live in Boulder of Boulder county, Colorado. User preference vector: [-0.05, -0.01, -0.02, -0.05, -0.00].
Recommended movie 1
Reward: 5

Context: This person is a 32-year-old man, working as a engineer and live in Little canada of Ramsey county, Minnesota. User preference vector: [-0.04, -0.00, -0.02, -0.04, -0.00]. Recommended movie 1 Reward: 4

You have a new user:

Context: This person is a 24-year-old woman studying as a student and live in Huntsville of Madison county, Alabama. User preference vector: [-0.04, 0.03, 0.05, -0.02, -0.01].

Which movie (id) should be recommended next? Show your reasoning between <think> </think> tags and the recommended movie index between <answer> </answer> tags, for example: <answer> movie 3 </answer> .

---

---

**RL with Algorithmic Reward (7B Gaussian; Best Checkpoint)**

In a 5-armed bandit problem, here are the results of previous arm pulls:

Arm 0: 2 pulls, average reward -1.421
Arm 1: 60 pulls, average reward 1.110
Arm 2: 2 pulls, average reward -2.903
Arm 3: 3 pulls, average reward -1.197
Arm 4: 1 pulls, average reward -1.842

Which arm should be pulled next? Show your reasoning in <think> </think> tags and your final answer in <answer> </answer> tags.

- - - - - - - - - - - - - - - - - - - - - - - - - - - - - - - - - - - - - - - - - -

<think> To decide which arm to pull next, we can use the Upper Confidence Bound (UCB) algorithm, which balances exploration and exploitation. UCB selects the arm with the highest upper confidence bound, which is the sum of the estimated mean reward and an exploration term. The formula for UCB is: $\text{UCB} = \text{average reward} + C \times \sqrt{\frac{\log(n)}{n}}$ where ( $n$ ) is the number of times the arm has been pulled, and ( $C$ ) is a tuning parameter typically set to 1 or 2. Let's calculate the UCB for each arm using ( $C = 1$ ):

For Arm 0: $\text{UCB}_0 = -1.421 + 1 \times \sqrt{\frac{\log(2+1)}{2}} \approx -1.421 + 0.549 \approx -0.872$

For Arm 1: $\text{UCB}_1 = 1.110 + 1 \times \sqrt{\frac{\log(60+1)}{60}} \approx 1.110 + 0.040 \approx 1.150$

For Arm 2: $\text{UCB}_2 = -2.903 + 1 \times \sqrt{\frac{\log(2+1)}{2}} \approx -2.903 + 0.549 \approx -2.354$

For Arm 3: $\text{UCB}_3 = -1.197 + 1 \times \sqrt{\frac{\log(3+1)}{3}} \approx -1.197 + 0.302 \approx -0.895$

For Arm 4: $\text{UCB}_4 = -1.842 + 1 \times \sqrt{\frac{\log(1+1)}{1}} \approx -1.842 + 0.693 \approx -1.149$

Now, we compare the UCB values:
Arm 0: -0.872
Arm 1: 1.150
Arm 2: -2.354
Arm 3: -0.895
Arm 4: -1.149
The highest UCB value is for Arm 1, which suggests we should pull Arm 1 next. </think> <answer> The arm that should be pulled next is Arm 1. </answer>

---

---

**RL with Algorithmic Reward (7B Bernoulli; Best Checkpoint)**

In a 5-armed bandit problem, here are the results of previous arm pulls:

Arm 0: 19 pulls, average reward 0.541
Arm 1: 1 pulls, average reward -0.879
Arm 2: 1 pulls, average reward -2.651
Arm 3: 46 pulls, average reward 1.240
Arm 4: 1 pulls, average reward -1.374

Which arm should be pulled next? Show your reasoning in <think> </think> tags and your final answer in <answer> </answer> tags.

- - - - - - - - - - - - - - - - - - - - - - - - - - - - - - - - - - - - - - - - - - - - - - - -

<think> To determine which arm to pull next, we can use the Upper Confidence Bound (UCB) algorithm, which balances exploration and exploitation by considering both the average reward and the uncertainty associated with each arm. The formula for UCB is: UCB = Average Reward + Exploration Term

For simplicity, we can use a basic form where the exploration term is inversely proportional to the square root of the number of pulls: UCB = Average Reward + $\frac{k}{\sqrt{\text{Number of Pulls}}}$ where ( k ) is a constant, typically set to 1 for simplicity.

Let's calculate the UCB for each arm:

Arm 0: $\text{UCB}_0 = 0.541 + \frac{1}{\sqrt{19}} \approx 0.541 + 0.229 \approx 0.770$

Arm 1: $\text{UCB}_1 = -0.879 + \frac{1}{\sqrt{1}} \approx -0.879 + 1 \approx 0.121$

Arm 2: $\text{UCB}_2 = -2.651 + \frac{1}{\sqrt{1}} \approx -2.651 + 1 \approx -1.651$

Arm 3: $\text{UCB}_3 = 1.240 + \frac{1}{\sqrt{46}} \approx 1.240 + 0.120 \approx 1.360$

Arm 4: $\text{UCB}_4 = -1.374 + \frac{1}{\sqrt{1}} \approx -1.374 + 1 \approx -0.374$

Now, we compare the UCB values: $\text{UCB}_0 \approx 0.770$
$\text{UCB}_1 \approx 0.121$
$\text{UCB}_2 \approx -1.651$
$\text{UCB}_3 \approx 1.360$
$\text{UCB}_4 \approx -0.374$.

Clearly, Arm 3 has the highest UCB value, indicating it is the best choice according to the UCB algorithm. </think>
<answer> The arm that should be pulled next is Arm 3. </answer>

---

**RL with Algorithmic Reward (7B Gaussian; Iteration 50 Checkpoint)**

In a 5-armed bandit problem, here are the results of previous arm pulls:

Arm 0: 2 pulls, average reward -0.282
Arm 1: 10 pulls, average reward 1.723
Arm 2: 5 pulls, average reward 1.255
Arm 3: 2 pulls, average reward -1.101
Arm 4: 2 pulls, average reward 0.995

Which arm should be pulled next? Show your reasoning in <think> </think> tags and your final answer in <answer> </answer> tags.

- - - - - - - - - - - - - - - - - - - - - - - - - - - - - - - - - - - - - - - - - - -

<think> To decide which arm to pull next, we can use the Upper Confidence Bound (UCB) algorithm, which balances exploration and exploitation. UCB selects the arm with the highest upper confidence bound of the expected reward. The formula for UCB is: UCB = Average Reward $+ C \times \sqrt{\frac{\log \text{ total number of pulls}}{\text{number of times arm was pulled}}}$.

Here, ( C ) is a tuning parameter that controls the trade-off between exploration and exploitation. For simplicity, we can assume ( C = 1 ).

First, calculate the total number of pulls: $2 + 10 + 5 + 2 + 2 = 21$.

Now, apply the UCB formula to each arm:

Arm 0: $\text{UCB}_0 = -0.282 + 1 \times \sqrt{\frac{\log 21}{2}} \approx -0.282 + 0.956 = 0.674$.

Arm 1: $\text{UCB}_1 = 1.723 + 1 \times \sqrt{\frac{\log 21}{10}} \approx 1.723 + 0.302 = 2.025$.

Arm 2: $\text{UCB}_2 = 1.255 + 1 \times \sqrt{\frac{\log 21}{5}} \approx 1.255 + 0.509 = 1.764$.

Arm 3: $\text{UCB}_3 = -1.101 + 1 \times \sqrt{\frac{\log 21}{2}} \approx -1.101 + 0.956 = -0.145$.

Arm 4: $\text{UCB}_4 = 0.995 + 1 \times \sqrt{\frac{\log 21}{2}} \approx 0.995 + 0.956 = 1.951$.

Comparing these UCB values, Arm 1 has the highest upper confidence bound. </think>
<answer> The arm that should be pulled next is Arm 1. </answer>

Figure 23: An example generated by `RL-ALG` policy at iteration 50 (7B Gaussian)

# E    LLM USE DISCLOSURE

**Using LLMs to help with paper writing.**   Commercial LLMs were used to correct typos and grammar, suggest alternative phrasings, and provide insights on the clarity and readability.   All LLM-generated text was reviewed, edited, and approved by the human authors.

**Using LLMs as a research assistant.**   LLMs assisted with brainstorming experimental designs, suggesting analysis approaches, searching potentially relevant prior work, and producing code scaffolding and completion. The human authors provided the research context, validated the literature identified by LLMs, verfied all analysis and results, and adapted or often rewrote the LLM-generated content before inclusion.

