# OpenReview forum: "When Greedy Wins: Emergent Exploitation Bias in Meta-Bandit LLM Training"
_ICLR.cc/2026/Conference — ICLR 2026 Poster_

### Official Review · Reviewer_pbuB · 2025-10-28

**Soundness:** 4
**Presentation:** 4
**Contribution:** 4
**Rating:** 10
**Confidence:** 3

**Summary:**

This paper investigates how Supervised Fine-Tuning (SFT) and Reinforcement Learning (RL) training paradigms shape the exploration strategies of Large Language Models (LLMs) when solving Multi-Armed Bandit (MAB) tasks, treating the LLMs as meta-bandit agents.

I currently think this paper provides a lot of value to the "learning to learn" / "learning to explore" community. However, I raised several questions and based on the authors' rebuttal response, I reserve the right to change/update my score.

**Strengths:**

1. The paper has extremely clear writing
2. The authors disambiguated different training strategies and setups very well.
3. This paper introduced token-level reward derived from GAE, which improved upon prior work.
4. Each reward design is principled and reasonable -- incorporating all possible learning signals.
5. The paper is well-situated compared to prior works, investigating areas where prior works didn't, adding critical knowledge/understanding to the entire teaching LLM to explore space.

**Weaknesses:**

The paper doesn't have obvious flaws/weaknesses, but here is some nitpicking weaknesses:

1. As a field, the learning to explore community needs to move beyond toy setting. [1] proposed and [2] included the MovieLens environment, which is a good first step. Behaviors of RL vs SFT and whether we need $\pi^*$ (optimal exploration policy) need to be answered in slightly more realistic setting. Many algorithms (like SGD) might be flawed in constructed setups, but they work really well in practice. Even though I think this paper has enough value without exploring those more realistic settings, I would encourage the authors to extend the paper to cover those settings.

2. The conclusion from the experiment seems a bit unclear (see the question part).

[1] Nie, Allen, et al. "Evolve: Evaluating and optimizing llms for exploration." arXiv preprint arXiv:2410.06238 (2024).

[2] Schmied, Thomas, et al. "Llms are greedy agents: Effects of rl fine-tuning on decision-making abilities." arXiv preprint arXiv:2504.16078 (2025).

**Questions:**

Typo 1: In Figure 3, it says RL-STG, it should be RL-STR

1. RL-STR is not significantly better than RL-OG. Just comparing the average, in 3 out of 8 domains (Figure 3), it's better. It's worse in 3 out of 8. In 2 it's about the same? For people who might want to train exploration policy in a realistic setting, how would you give them suggestions on what reward design to use?

2. Table 2, you did some behavioral analysis and citing a prior work. Krishnamurthy et al. (2024) also proposed MinFrac. Do you think this is a metric worth including?

3. [1] also proposed to fit a function form over steps $T$ on average cumulative regret -- and try to describe the slope/speed of learning/unlearning. Would such analysis yield additional insights for your setting?

4. What future directions are opened up by your analysis, besides moving to more realistic domain? In your GRPO training (with RL-OG), do you see length hacking (i.e., with training continues, the length in thinking tokens becomes longer and longer)? In appendix, we saw that RL with ALG reward learns to compute UCB. Can you:
   - Provide sampled generations with base model but sampled using your prompt/pipeline? Do they already know that they should calculate UCB values?
   - Can you provide sampled generation from RL-OG and RL-STR for us so we know if these also learned (on their own) to compute UCB values?

---

> ### Author Response · Authors · 2025-11-22
> **Author Responses (1/n)**
>
> We thank the reviewer for their extremely positive reviews and insightful questions! We appreciate the opportunity to incorporate these suggestions and hope to address the specific questions below.
>
> First, we apologize for the typo of “RL-STR”, we have consistently used RL-STG (RL training with regret-based strategic rewards) throughout the paper, except for its initial introduction where the typo occurred. We will update the manuscript to unify all usage to RL-STG for consistency.
>
> ---
> > extension to the MovieLens environment
>
> We agree the contextual bandit environment could be a valuable addition and our proposed approach naturally generalizes to such settings. We focus on MAB for its controlled setup and well-defined challenges, which are often lacking in more realistic setups such as the MovieLens, as we will elaborate below.
>
> We conducted a series of experiments on the MovieLens environment, following the setup described in Nie et al, and Schmied et al. We use the Qwen 2.5 7B model as our base model and LinUCB (exploration ratio 0.5) as the baseline. Using hyper-parameters similar to those in our paper, we trained the agent over a horizon of 50 with 5 arms (recommending from 5 movies). We found that both RL with environmental feedback and RL with algorithmic reward reduce the regret compared to the Pretrain LLM. Compared to the finding in Schmied et al, we again show our multi-turn RL implementation can outperform strong baseline in terms of lower regret. RL-ALG performs worse than RL-OG because it struggles to follow the exact LinUCB calculation, which is even harder than UCB for MAB. Consequently, it again pursues a “greedy” approximation, converging to the action the LinUCB teacher chooses 86% of the time (i.e., the most popular movie).
>
> |  | Avg Cumulative Regret |
> | :--- | :--- |
> | LinUCB | $61.09( \pm 2.57)$ |
> | 7B Pretrain | $100.84( \pm 3.09)$ |
> | 7B RL-OG (full history) | $46.64( \pm 1.70)$ |
> | 7B RL-OG (no history) | $58.71( \pm 1.32)$ |
> | 7B RL-ALG | $73.23( \pm 1.78)$ |
>
> **Limitations of MovieLens for meta-bandit training**: Unlike the MAB setup, where we can generate infinite environments with different reward distribution parameterizations, MovieLens is fundamentally a single bandit instance defined by one static dataset. The ground-truth reward for any user $i$ and movie $j$ is fixed and deterministic: $r_{i,j} = u_i^T \Sigma v_j$. Since $u_i$ is given in the prompt, it essentially requires the modeling of a linear function between a user feature vector and a reward distribution over a fixed action set. Although unseen user features appear during testing, the action set is fixed and the linear function $ \Sigma$ is universal. This allows the model to memorize the function during our meta-bandit training, significantly undermining the online learning component core to bandits. Empirically, the RL agent achieved lower regret than LinUCB even without access to interaction history. Deviating from Nie et al's setup, one may use different movies as variable action sets across instances, which could restore the bandit challenge. However, the dataset is sparse and ratings are concentrated among popular movies, the noisy reward distribution, even if learnable, requires much longer horizon and long context LLM, which exceed our training resources.
>
> We will include these results and the discussion in the appendix.
>
> ---
> > RL-STG is not significantly better than RL-OG. .. in 3 out of 8 domains (Figure 3), it's better. It's worse in 3 out of 8.
>
> By design, RL-STG is significantly better than RL-OG specifically in environments with high-variance reward distributions (e.g., the four evaluations with a Gaussian env trained agent in Figure 3, row 1). In low-variance environments, the results are largely similar, especially for in-domain evaluation. The minor degradation in OOD results suggests potential overfitting to the training environment. We recommend using RL-STG when the underlying reward distribution is accessible during training. We will clarify this distinction in the manuscript.
>
> ---
> > MinFrac (Krishnamurthy et al.) and Function fitting (Nie et al.)
>
> We followed the suggestion to report these additional metrics. We observe that both SFT and RL significantly reduce the k*MinFrac@50 of Pretrain 7B LLM (from 35.2 to ~10). Function fitting shows that Pretrain LLM exhibits linear regret (similar to Greedy), while fine-tuning greatly reduces the linearity, reaching levels comparable to or lower to UCB. We will add MinFrac to Table 2 and the function fitting results to the appendix.
>
> | Metric | k*MinFrac |  |
> | :--- | ---: | ---: |
> | $@ t$ | 50 | 300 |
> |  |  |  |
> | Greedy | 10.2 | 1.7 |
> | $\epsilon$-Greedy | 2.5 | 8.5 |
> | TS | 15.2 | 4.0 |
> | UCB | 10.5 | 1.8 |
> |  |  |  |
> | Pretrain | 35.2 | 18.1 |
> | SFT | 10.5 | 1.8 |
> | RL-OG | 13.3 | 3.6 |
> | RL-STG | 10.9 | 2.4 |
> | RL-ALG | 10.0 | 1.7 |

---

> ### Author Response · Authors · 2025-11-22
> **Author Responses (2/n)**
>
> > do you see length hacking
>
> We don’t observe length hacking, the average length of generation usually stabilized after increasing or decreasing to a certain level. However, we do observe that sometimes a very small fraction of generated sequence could exceed the maximum token limit. In such cases, the RL training is susceptible to collapse (length explosion). A format reward and resampling when exceeding length can sufficiently alleviate this. We leave the in-depth analysis of this LLM RL instability issue to future work.
>
> ---
> > Do [LLMs] already know that they should calculate UCB values?
>
> Yes, our analysis suggests this knowledge exists in the pretrain parameters. As discussed in the last paragraph of section 5 and appendix C, the LLM can generate solutions with correct UCB formulation in early iteration. However, these correct strategies are often discarded as the model converges, likely due to failures in credit assignment. For example, even if the LLM generates the correct UCB formulation, it may receive a low reward due to calculation errors involving square root and logarithm. It’s harder to improve these basic arithmetic abilities with sparse outcome rewards than to adopt a greedy strategy that is easier to implement and matches the teacher’s decision most of the time.
>
> ---
> >  if RL-OG and RL-STG also learned (on their own) to compute UCB values?
>
> Yes, as such knowledge is already in the pretrain model, these two RL agents also often (about 25% of the times) attempt to calculate UCB values. We will surely include more generation samples in the appendix to demonstrate this.
>
> ---
>
> Lastly, we note that prior research on LLM bandits from industrial labs didn’t release training code (Nie et al. did release their valuable eval benchmark but not training). We are committed to releasing our code and eval for reproductivity and further investigation of multi-turn LLM RL in interactive environments, which embody immense engineering challenges. We truly appreciate your support in helping us contribute to open science.

---

### Official Review · Reviewer_CjPh · 2025-11-01

**Soundness:** 3
**Presentation:** 3
**Contribution:** 2
**Rating:** 2
**Confidence:** 4

**Summary:**

This submission studies how LLMs learn to balance exploration and exploitation when trained to solve multi-armed bandit (MAB) problems, using both supervised fine-tuning and reinforcement learning. They train LLMs on various MAB environments using RL with three forms of reward design: original reward (i.e. the raw environment rewards), strategic reward (a normalized, regret-shaped reward that reflects how close the agent’s action is to the optimal arm at each tilmestep), and algorithmic reward (a signal that rewards the agent for matching the decisions of an expert oracle, specifically the UCB algorithm). For supervised fine-tuning, they train the LLM on synthetic data generated from UCB expert trajectories, including step-by-step chain-of-thought demonstrations of UCB value calculations.

They evaluate the performance of these methods (when fine-tuning Qwen models) on Gaussian and Bernoulli MAB environments. They find that the fine-tuned models outperform vanilla models on these tasks, and that policies trained with UCB imitation perform best. These methods also perform comparably to UCB and Thompson sampling.

Upon further analysis, the authors find that these gains often come from the models adopting greedy behaviors and under-exploring.

**Strengths:**

Using LLMs to solve decision-making tasks under uncertainty is an important next frontier, so the problem being studied is well-motivated. While the authors are not the first to fine-tune LLMs to solve these sorts of bandit tasks, their proposed reward designs are novel, from what I can tell. The finding that gains often stem from more greedy behavior is also interesting, and largely matches the findings of Krishnamurthy et al for vanilla models which are not fine-tuned.

**Weaknesses:**

I am not entirely sure what this paper is trying to accomplish. The typical motivation for studying LLMs’ ability to solve bandit tasks is because we want to design AI models capable of solving complex, text-based, decision making tasks under uncertainty. This is because we don’t currently have good algorithms for solving these types of tasks. Bandits get to the essence of the exploration/exploitation trade-off, and so they can be a useful abstraction.

However, I am not sure what utility we get from fine-tuning LLMs to be more like existing algorithms, without evaluating their performance on some complex task that existing algorithms do not do well in (or are not even applicable to). (We don't need to reinvent the wheel to solve MAB tasks, as we have good algorithms for these types of tasks already.)

**Questions:**

What are you trying to show when evaluating these fine-tuned models on MAB tasks? It is not very surprising that a transformer will behave like UCB if you train it to.

---

> ### Author Response · Authors · 2025-11-22
> **Author Responses**
>
> We thank the reviewer for recognizing the technical novelty of our algorithm designs and the interestingness of our findings. We appreciate the opportunity to clarify the motivation behind our experimental setting.
>
> ---
> > I am not sure what utility we get from fine-tuning LLMs to be more like existing algorithms, without evaluating their performance on some complex task that existing algorithms do not do well in
>
> Our main goal in the imitation learning experiments (namely RL-ALG and SFT) is **not** to engineer a state-of-the-art meta-bandit agent, but rather to isolate and study the ubiquitous emergence of exploitation bias during task fine-tuning across training setups. By using an optimal teacher we remove the difficulty of stochastic rewards and long-horizon credit assignment across bandit turns. We observe that the exploitation bias emerges even in this simplified supervision setting. This suggests that this bias isn’t just a result of noisy RL feedback, but a fundamental property of the task fine-tuning dynamics. This validates our findings from multi-turn RL settings (i.e., RL-OG and RL-STG) which solely learn from environmental feedback.
>
> Moreover, for powerful LLMs with strong pre-training priors, imitation learning (namely RL-ALG) acts as behavioral shaping rather than strict replication. The teacher restricts the student’s final actions, but the LLM generates the solution using its own internal reasoning (e.g., CoT). In our MAB experiments, the student failed to internalize the strict mathematical logic of the UCB teacher, instead discovering flawed greedy variants. This counter-intuitively resulted in the student exceeding the teacher's performance (lower average regret). This finding suggests that strong LLMs can potentially learn from accessible, weak teachers and outperform them by optimizing execution via their own priors.
>
> ---
> > What are you trying to show when evaluating these fine-tuned models on MAB tasks?
>
> We employ the classic abstraction of MAB not as the final application but as a sanity check or **necessity test** for complex sequential decision-making. It isolates the exploration-exploitation trade-off with controlled environments that avoid potential confounding factors or shortcuts of realistic tasks. Our findings that both imitation learning and task RL induce greedy behaviors hidden behind conventional regret analysis serve as a crucial warning.
>
> Moreover, our proposed training framework and approaches are generalizable to other learning to explore setups. We have extended our experiments to the MovieLens environment, a widely used contextual bandit benchmark for LLM agents proposed by [Nie et al. 2025](https://arxiv.org/abs/2410.06238). Our RL fine-tuning (RL-OG) successfully outperforms strong baselines (including the Pretrained LLM and LinUCB), achieving lower regret. Interestingly, we observed that the "greedy" behavior persists: the RL-ALG agent converges to the most popular movie (it was also chosen by the LinUCB teacher 86% of the times), effectively using a greedy approximation to bypass the complex linear algebra required for exact LinUCB calculation, which in this case causing higher regret than the teacher.
>
> |  | Avg Cumulative Regret |
> | :--- | :--- |
> | LinUCB | $61.09( \pm 2.57)$ |
> | 7B Pretrain | $100.84( \pm 3.09)$ |
> | 7B RL-OG | $46.64( \pm 1.70)$ |
> | 7B RL-ALG | $73.23( \pm 1.78)$ |
>
> ---
> > It is not very surprising that a transformer will behave like UCB if you train it to.
>
> We respectfully point out that the LLMs behave **differently** to UCB, despite being trained to do so. In the abstract, we state that “agents trained to imitate UCB learn to outperform their teacher by adopting more exploitative variants”. This deviation is investigated in section 5.2, for example Figure 5 showcases the students opt for different actions that can be better described by flawed UCB variants. The students systematically amplify the teacher’s greedy tendency while failing to mimic its complex exploration mechanisms (i.e., UCB calculations). This bias amplification is a novel insight into LLM behavior during task fine-tuning as opposed to pretraining (Krishnamurthy et al).
>
> Thanks again for the valuable feedback, if you feel our response has helped bring in new perspectives, we would be grateful if you would reconsider your score!

---

> > ### Comment · Reviewer_CjPh · 2025-11-26
> >
> > Thanks for your reply. I agree that the observation that fine-tuned LLMs act "too greedy" is an interesting one, but it is not very surprising given that Krishnamurthy et al show that the base models also often behave too greedily. On this note, I'm not sure it is fair to say that the greedy behavior "emerges" during task fine-tuning, as it has likely been there from the start.
> >
> > Could you please provide more details about your new experiments on MovieLens? (Either here or by uploading a new version of the submission.)

---

> ### Author Response · Authors · 2025-12-03
> **Author Responses**
>
> We thank the reviewer for the engagement and the acknowledgment of the *interestingness* of our findings. We appreciate the opportunity to clarify these final points.
>
> ---
> > more details about your new experiments on MovieLens
>
> We have added the detailed setup and results regarding the contextual bandit experiments on MovieLens to Appendix D of the updated manuscript.
>
> ---
> > it is not very surprising given that Krishnamurthy et al show that the base models also often behave too greedily
>
> While Krishnamurthy et al. demonstrate that frontier models (e.g., GPT-4) can behave greedily or uniformly depending on the prompting setup, their study analyzes the **outcome** of undisclosed training on black-box models where the contribution of training is impossible to figure out.
>
> Our work instead investigates the **process** of learning in-context exploration under controlled task-specific SFT and RL paradigms. When we describe behavior as "emergent," we are not referring to generic greediness, but to specific structural behavioral shifts that are absent in our base model but appear distinctively after specific training. In abstract, we frame it as "more sophisticated but greedier exploitation". We present comprehensive evidence, including
> - In Figure 5, the models shift from a prior normal-like distribution to a distinct bimodal distribution only after training. (we moved this figure from appendix to the main text)
> - In Table 2, the suffix failures emerge only after the fine-tuning process. Imitation learning policies amplify the teacher's tendency.
> - In Section 5.2, detailed analysis into flawed variants of UCB in the CoT of RL-ALG and SFT agents.
>
> Therefore, our contribution is not merely observing that LLMs can be greedy, but rather mechanistically showing how different learning signals (SFT vs. RL with various rewards) drive the model toward specific types of greedy exploitation and failure modes that were not present before the task-specific fine-tuning.
>
> We hope this further clarifies our contribution.

---

### Official Review · Reviewer_USow · 2025-11-03

**Soundness:** 2
**Presentation:** 2
**Contribution:** 3
**Rating:** 6
**Confidence:** 4

**Summary:**

This paper studies how different fine-tuning paradigms shape the exploration and exploitation behaviors of LLMs in multi-armed bandit (MAB) settings. The authors train Qwen-2.5 3B and 7B models on MAB tasks with different reward signals:
> RL-OG: standard stochastic rewards,

> RL-STG: regret-shaped strategic rewards, and

> RL-ALG: imitation of the UCB oracle via algorithmic rewards.

The work demonstrates that both SFT and RL reduce regret and achieve performance comparable to theoretical baselines like UCB and Thompson Sampling, with generalization to unseen environments and 6$\times$ longer horizons. However, behavioral analysis shows that these gains arise from emergent exploitative tendencies since trained agents become greedier and sometimes abandon exploration prematurely. RL-ALG agents trained to imitate UCB outperform their teacher by adopting more exploitative variants of it.
The paper concludes that while both SFT and RL enhance decision-making, they also introduce bias toward short-term exploitation, highlighting the need for reward and data designs that explicitly sustain exploration.

**Strengths:**

- **Originality and significance**: This work provides a unified framework comparing SFT and RL fine-tuning for LLM agents on the same controlled MAB setup. The strategic and algorithmic rewards are well-motivated and address variance and credit-assignment issues in RL with LLMs.
- **Quality**: This paper includes experiments across Gaussian and Bernoulli bandit families, including cross-distribution generalization and longer horizons, which strengthen the empirical claims. Metrics like suffix failure and greedy frequency are used to reveal qualitative behavioral shifts alongwith quantitative evaluation.
- **Clarity**: The paper is generally well-written with clear implementation details and code release commitments.

**Weaknesses:**

- Limited task diversity: Experiments are confined to simple bandit environments, therefore claims about general agentic learning remain somewhat speculative.

- Model scalability: There is no discussion of whether the observed training and inference dynamics hold even with larger models and across different model families.

- While the paper uncovers behavioral tendencies, it stops short of offering causal or mechanistic explanations of why greediness emerges under RL/SFT objectives.

- Compared to the analysis of RL approaches in LLM training, the SFT analysis is less thorough, especially regarding data sampling effects and robustness to arithmetic errors.

- No explicit exploration-promoting baseline: Methods like intrinsic reward shaping or information gain are not compared, though they would provide valuable context.

**Questions:**

Exploration bias mitigation – Could future training include explicit exploration bonuses (e.g., curiosity or entropy regularization) to test if the emergent greediness persists?

Do the authors have an intuition of how findings might transfer to contextual or Markovian tasks where exploration requires memory and state estimation?

The finding that RL-ALG outperforms its UCB teacher is quite interesting. Could the authors analyze whether this over-exploitation emerges from token-level credit assignment or from the episodic reward structure?

---

> ### Author Response · Authors · 2025-11-22
> **Author Responses**
>
> We thank the reviewer for their positive review and interesting questions! We address the key concerns below:
>
> ---
> > “Limited task diversity”, “How findings might transfer to contextual... tasks”
>
> We focused our experiments on MAB as a clean, controllable abstraction and a sanity test. It isolates the exploration-exploitation challenge without the confounding variables of complex state estimation found in MDPs. If LLMs struggle to learn exploration strategies in MAB, they will fundamentally struggle in more complex agentic environments that embody the same challenge.
>
> To provide more empirical evidence regarding generalization, we have extended our experiments to the MovieLens environment, a widely used contextual bandit benchmark for LLM agents proposed by [Nie et al. 2025](https://arxiv.org/abs/2410.06238). Our RL fine-tuning (RL-OG) successfully outperforms strong baselines (including the Pretrained LLM and LinUCB), achieving lower regret. Interestingly, we observed that the "greedy" behavior persists: the RL-ALG agent converges to the most popular movie (it was also chosen by the LinUCB teacher 86% of the times), effectively using a greedy approximation to bypass the complex linear algebra required for exact LinUCB calculation.
>
> |  | Avg Cumulative Regret |
> | :--- | :--- |
> | LinUCB | $61.09( \pm 2.57)$ |
> | 7B Pretrain | $100.84( \pm 3.09)$ |
> | 7B RL-OG | $46.64( \pm 1.70)$ |
> | 7B RL-ALG | $73.23( \pm 1.78)$ |
>
> ---
> > Why RL-ALG outperforms its UCB teacher ("Analyze whether this over-exploitation emerges from token-level credit assignment or from the episodic reward structure...")
>
> This is an interesting aspect of our findings. Our results suggest that it’s a combination of both factors.
> - Firstly, compared to the RL-ALG approach, the SFT agent is able to match the teacher at a higher accuracy, and its greedy tendency is weaker when evaluated in-distribution (Table 2). This shows dense supervision can alleviate the exploitation bias, which can be achieved by better token-level credit assignment in RL setup.
> - Secondly, regarding the episodic reward structure, we observe that training RL-ALG agents on environments that require more exploration alleviates their greedy tendency. For example, the UCB algorithm adopts 3x fewer greedy actions in `Bernoulli5_Uniform` compared to `Gaussian5_Var1_MeanN0`. Consequently, the RL-ALG agent trained on the `Bernoulli5_Uniform` consistently has lower GreedyFreq than the one trained on the `Gaussian5_Var1_MeanN0`. This suggests that the episodic reward structure of the training env influences the RL-ALG agent’s behavior.
>
> | train↓ \ eval→  | Bernoulli | Gaussian |
> | :--- | :--- | :--- |
> | Bernoulli | 74.9 & 85.3 | 84.4 & 95.6 |
> | Gaussian | 78.2 & 91.9 | 85.4 & 97.0 |
>
> Table: GreedyFraq of 7B RL-ALG agents at 50 and 300 steps, the first row is trained on the `Bernoulli5_Uniform` environment, and the second row is trained on the `Gaussian5_Var1_MeanN0` environment.
>
> We will include the above discussion into the manuscript to clarify this point.
>
> ---
> > the SFT analysis is less thorough, especially regarding data sampling effects and robustness to arithmetic errors
>
> We discuss in the paper the arithmetic errors of the SFT agent can hinder their generalization to unseen negative rewards and longer horizons. Figure 5 and corresponding paragraphs illustrate the calculation errors of the UCB values. In the appendix C, we show that lowering the environment’s mean reward leads to more unseen negative rewards and triggers more imitation errors. The data for SFT is sampled uniformly at episode level, given a fixed teacher target, there is no trivial way to directly adjust the sampling distribution. Overall, we prioritize the analysis of RL agents, as they are more applicable to multi-turn decision making tasks compared to SFT. However, with the additional page allowance, we will move the analysis of SFT from appendix to the main paper.
>
> ---
> > Could future training include explicit exploration bonuses (e.g., curiosity or entropy regularization) to test if the emergent greediness persists?
>
> We agree, and we have listed these as promising future directions in our conclusion section. Our preliminary experiments with information gain-based reward shaping and token-level entropy regularization weren’t successful. These approaches require sophisticated designs and hyper-parameter tuning which is beyond our main contribution; we leave them to future work.
>
> We hope our response has addressed your concern and provided a better understanding of our contribution. We would appreciate it if you would consider raising your score.

---

### Comment · Area_Chair_ZMt8 · 2025-11-27
**Reviewer Reminder: Author Rebuttals Available**

Dear Reviewers,

The authors have posted their rebuttals to your reviews.

Please read the authors' responses, assess whether your concerns have been addressed, and update your rating and confidence accordingly.

Your prompt attention to the rebuttals is appreciated.

Best,
AC

---

### Author Response · Authors · 2025-12-03
**General Response to Reviewers**

We thank all reviewers for their constructive feedback and the time spent engaging with our work. We appreciate that **all** reviewers acknowledged our methodological soundness and novelty, and found our analysis regarding the emergent "greedy" behavior interesting. Especially, reviewers `USow` and `pbuB` positively supported the significance of our work. Our initial response to Reviewer `CjPh` convinced them to recognize the interestingness of the emergent behavior, and we hope our further response has clarified the remaining confusion (unfortunately ICLR had to prematurely end the discussion period this year).

We have incorporated the reviewers' suggestions into the updated manuscript. We believe these changes have significantly strengthened our paper:

* Generalization to contextual bandits (Appendix D): Addressing the request for more realistic settings (Reviewers `USow` & `pbuB`), we have added a new section detailing experiments on the MovieLens environment. This demonstrates the generalizability of our RL approach, while more controllable MAB environments remain our focus.
* Additional metrics (Table 2 & Figure 8): Per Reviewer `pbuB`'s suggestion, we have included MinFrac and function fitting analyses. These metrics provide consistent findings of the learning dynamics and confirm that fine-tuning significantly improve regret in-average compared to pretrained models.
* Action space generalization (Figure 5): We added the figure to demonstrate that our training fundamentally improves the generalization of the policy to larger action spaces, a setting where the pretrained model typically fails.
* Visualizing behavioral shifts (Figure 6): We moved the histogram analysis (previously in the appendix) to the main text, with visual revisions. This illustrates the specific structural behavioral shift that emerges specifically *after* task-specific fine-tuning.
* Misc: We have fixed minor errors, including the typo pointed out by the reviewers.

We thank the reviewers again for helping us improve the quality of this manuscript.

---

### Meta-Review · Area_Chair_SDve · 2026-01-05

**Summary:**

This paper investigates the impact of different approaches in enhancing the exploration capabilities of LLMs. It does so by training LLMs with three different reward functions: (1) the original reward, a regret-shaped reward, and an imitation reward. Overall, this paper has been quite polarizing, receiving ratings of 2 and 10. I think it is clear that the paper is not trying to propose LLMs as an alternative to MAB, but it is using MAB as a proxy to examine the exploratory capabilities of these LLMs. In this context, the real question is whether the results are interesting or not. All reviewers seem to agree that those results are interesting. Thus, given that the paper got someone so excited that they would give the paper a 10, I'd rather not gatekeep this paper, and I am recommending its acceptance.

**Reviewer Concerns:**

- _Experiments are confined to simple bandit environments._

	The authors did a good job isolating a scientific question in a controllable environment, so I don't see this as a weakness.

- _Would the observed training and inference dynamics hold even with larger models and across different model families?_

	One can always ask that, but I don't think it would invalidate what is being reported in the paper.

- _Isn't it obvious that an LLM will behave more like an algorithm able to explore when it is trained to do so? And isn't it obvious that it will be more greedy in light of previous related work?_

	I think this is the real criticism and it is the one that I find hard to dismiss. Clearly, the goal of the authors is not to solve MAB problems, though, as there are algorithms for that. Thus, they are using MAB as a proxy task to evaluate the performance of LLMs when exploring. In that context, the real question is about how interesting the observed phenomenon actually is in light of the fine-tuning that was performed.

**Reviewer Scores:**

- Reviewer USow: Initial rating: 6. I suspect the reviewer would have kept his rating the same, given that few new results were introduced and the authors argued against some of the raised weaknesses, but in my experience, this rarely moves reviewers.

- Reviewer CjPh: Initial rating: 2. The reviewer even responded to the author's rebuttal, and I find it unlikely they would raise their scores from a 2.

- Reviewer pbuB: Initial rating: 10. They wouldn't lower their score.

---

### Decision · Program_Chairs · 2026-01-26

Accept (Poster)